# Reduced Siderite Ore Combined with Magnesium Oxide as Support Material for Ni-Based Catalysts; An Experimental Study on CO$_2$ Methanation

Kamonrat Suksumrit [1], Christoph A. Hauzenberger [2], Srett Santitharangkun [2] and Susanne Lux [1,*]

[1] Institute of Chemical Engineering and Environmental Technology, Graz University of Technology, NAWI Graz, Inffeldgasse 25C, 8010 Graz, Austria; k.suksumrit@tugraz.at

[2] Department of Earth Sciences—NAWI Graz Geocenter, University of Graz, Universitätsplatz 2, 8010 Graz, Austria; christoph.hauzenberger@uni-graz.at (C.A.H.); srett.santitharangkun@edu.uni-graz.at (S.S.)

* Correspondence: susanne.lux@tugraz.at

**Abstract:** Ni-based catalysts play a fundamental role in catalytic CO$_2$ methanation. In this study, the possibility of using siderite ore as a catalyst or catalytic support material for nickel-based catalysts was investigated, aiming at the exploitation of an abundant natural resource. The catalytic performance of Ni-based catalysts with reduced siderite ore as a support was evaluated and compared to MgO as a support material. MgO is known as an effective support material, as it provides access to bifunctional catalysts because of its basicity and high CO$_2$ adsorption capacity. It was shown that undoped and Ni-doped reduced siderite ore have comparable catalytic activity for CO$_2$ hydrogenation (20−23%) at 648 K, but show limited selectivity toward methane (<20% for siderite$_{reduced}$ and 60.2% for Ni/siderite$_{reduced}$). When MgO was added to the support material (Ni/siderite$_{reduced}$/MgO), both the CO$_2$ conversion and the selectivity toward methane increased significantly. CO$_2$ conversions were close to the thermodynamic equilibrium, and methane selectivities of ≥99% were achieved.

**Keywords:** CO$_2$ methanation; Ni-based catalyst; magnesium oxide; siderite ore

## 1. Introduction

Carbon dioxide (CO$_2$) is one of the major contributors to global climate change. To mitigate CO$_2$ emissions, increasing attention is directed to carbon capture and utilization (CCU) and carbon utilization (CU) technologies. These technologies aim at utilizing CO$_2$ from industrial processes as raw material for the production of value-added chemicals and fuels, for instance, methane (CH$_4$) [1–3], methanol (CH$_3$OH), and gasoline [4].

Methane synthesis by CO$_2$ hydrogenation has gained wide attention from researchers all over the world [5–7]. Methanation, referring to the conversion of carbon monoxide (CO) and/or carbon dioxide with hydrogen (H$_2$) into methane, is thermodynamically favorable (Equations (1) and (2)). Both reactions are highly exothermic; thus, high temperatures are unfavorable for the carbon oxide conversion [3,8,9].

$$CO_2 + 4\,H_2 \rightleftharpoons CH_4 + 2\,H_2O \quad \Delta_R H^0 = -165.0 \text{ kJ mol}^{-1},\ \Delta_R G^0 = -113.61 \text{ kJ mol}^{-1} \quad (1)$$

$$CO + 3\,H_2 \rightleftharpoons CH_4 + H_2O \quad \Delta_R H^0 = -206.2 \text{ kJ mol}^{-1},\ \Delta_R G^0 = -142.25 \text{ kJ mol}^{-1} \quad (2)$$

Hydrogenation of CO$_2$ may also result in CO formation via the reverse water-gas shift reaction (Equation (3)).

$$CO_2 + H_2 \rightleftharpoons CO + H_2O \quad \Delta_R H^0 = -41.2 \text{ kJ mol}^{-1},\ \Delta_R G^0 = -28.64 \text{ kJ mol}^{-1} \quad (3)$$

To catalyze CO$_2$ methanation, ruthenium (Ru)- and rhodium (Rh)-based catalysts have shown promising catalytic activity and selectivity [10,11]. However, their high costs are

disadvantageous for industrial application. Currently, nickel (Ni)-based catalysts are the most widely used catalysts for $CO_2$ methanation because of their high catalytic activity, selectivity toward methane, and long-term stability. They help overcome the activation energy barrier, allowing the reaction to occur under mild conditions [12,13]. In addition, nickel is an abundant, relatively low-cost metal that helps extend the operational lifetime of the catalyst and thus reduces costs for catalyst replacement, which is crucial for industrial application. It is evident that the support material for the catalytically active nickel species has a pronounced effect on the activity and selectivity of the catalyst [14–18]. A series of different support materials for Ni-based catalysts is presented in the literature. Aluminum oxide ($Al_2O_3$), silicon dioxide ($SiO_2$), titanium dioxide ($TiO_2$), cerium oxide ($CeO_2$), and zirconium dioxide ($ZrO_2$), along with magnesium oxide (MgO), have been reported as support materials for Ni-based methanation catalysts [19–27]. Rahmani et al. produced a range of Ni catalysts supported on mesoporous nanocrystalline $\gamma$-$Al_2O_3$. The catalysts possessed large surface areas, with the 20 wt% Ni/$Al_2O_3$ catalyst showing the highest activity and stability between 473 K and 623 K [28]. Xu et al. investigated $CO_2$ methanation using Ni/$SiO_2$ catalysts that were prepared through a combustion-impregnation method and obtained a $CO_2$ conversion of 66.9% and a methane selectivity of 94.1% at 593 K [29]. However, the challenges encountered when using alumina and silica as support materials were carbon deposition and poor catalyst stability at the reaction temperatures [27,30]. Perkas et al. [31] developed Ni catalysts supported on mesoporous $ZrO_2$ modified with Ce and Sm cations, featuring 30 mol% Ni loading. These catalysts exhibited elevated catalytic activity for $CO_2$ methanation with a turnover frequency of 1.5 s$^{-1}$ at 573 K. Ni/$ZrO_2$ catalysts with various amounts of tetragonal polymorph $ZrO_2$ were prepared from an amorphous Ni-Zr alloy by Yamasaki et al. [32]. The tetragonal zirconia-supported nickel nanoparticles showed an even higher turnover frequency (TOF = 5.43 s$^{-1}$ at 473 K). Nevertheless, $ZrO_2$ and $CeO_2$ are expensive compared with other widely used support materials. Therefore, MgO has become an attractive alternative support material. Apart from being cost-effective, MgO has the significant advantages of exhibiting increased basicity of the surface, preventing catalyst deactivation, and mitigating issues of sintering and the formation of carbon deposits. Several studies have shown the efficiency and potential of MgO as a support material [33–35]. Ho et al. examined the $CO_2$ adsorption capability of MgO at various temperatures from 303 to 623 K [36]. As expected, with increasing temperatures, the $CO_2$ uptake capacity decreased. Takezawa et al. studied 13% Ni/MgO catalysts for $CO_2$ methanation. The catalysts were prepared by impregnation and calcination at temperatures of 673−973 K. The study revealed that with increasing calcination temperatures, the activity and selectivity of the Ni/MgO catalysts decreased. A methane selectivity of 98% was presented for catalyst calcination at 773 K followed by reduction at 873 K, a reaction temperature of 480 K, a pressure of 1 atm, and a feed gas flow rate of 10 mL min$^{-1}$ ($CO_2$:$H_2$ = 5:95) [37]. Varun et al. prepared NiO/MgO nanocomposite catalysts for $CO_2$ hydrogenation via sonochemical treatment and achieved a $CO_2$ conversion of 85%, with 98% selectivity toward methane at 673 K [38]. Baldauf-Sommerbauer et al. investigated two catalysts with 11 and 17 wt% Ni on MgO. The $CO_2$ conversion and methane selectivity approached the thermodynamic equilibrium at a moderate reaction temperature of 598 K and a feed composition of $H_2$:$CO_2$:$N_2$ = 4:1:5 at a feed gas flow rate of 250 mL$_{STP}$ min$^{-1}$ [39]. Loder et al. investigated the effect of the Ni loading (0−27 wt%) on MgO and the MgO quality on the rate of $CO_2$ methanation in a temperature range of 533–648 K. They reported $CO_2$ conversions of 87% and a methane selectivity of $\geq$99% [9]. In a current review, the role of carbonate formation during $CO_2$ hydrogenation over MgO-supported catalysts was discussed, explicitly stating the beneficial effect of bifunctional Ni/MgO catalysts toward methane synthesis [40]. In the case of Ni/MgO catalysts, nickel provides the adsorbent capacity for hydrogen and is highly selective for methane, whereas the basic support material, MgO, activates $CO_2$ through chemisorption, giving access to a highly active bifunctional catalyst.

In addition, Pandey et al. showed that adsorbed carbonate species on iron oxide sites on Ni-Fe catalysts serve as additional factors that enhance the efficiency of catalysts for $CO_2$ hydrogenation. The presence of a Ni-Fe alloy and a number of metal sites were reported to enhance $CO_2$ conversion and methane yield [41]. In general, iron-based catalysts were suggested as a cost-effective alternative for methane synthesis from $CO_2$, providing long-term stability when appropriately combined and modified with promoters [42]. Sehested et al. confirmed that a Ni-Fe alloy catalyst gave higher $CO_2$ conversion and $CH_4$ selectivity compared with a pure nickel catalyst at 603 K with excess hydrogen [43]. Mutz et al. studied the potential of $Ni_3Fe$ on $\gamma$-$Al_2O_3$ as a methanation catalyst in a microchannel packed bed reactor. With a 17% $Ni_3Fe$ catalyst, a $CO_2$ conversion of 71% and a selectivity toward methane > 98% were achieved at 631 K and 6 bar in long-term experiments (45 h). Thus, the $Ni_3Fe$ catalyst showed outstanding performance and stability at mid-temperatures when combining Ni and Fe [44]. In addition, Serrer et al. investigated bimetallic $Ni_{3.2}Fe/Al_2O_3$ catalysts by using an advanced combination of operando XAS and XRD with quantitative on-line product analysis. The results showed that Fe addition to $Ni/Al_2O_3$ catalysts protected active $Ni^0$ species from oxidation and preserved the catalytic activity under dynamic reaction conditions [45]. Mebrahtu et al. studied Ni-Fe bimetallic catalysts on a $(Mg,Al)O_x$ support, which were synthesized by co-precipitation. Ni-Fe alloy nanoparticles were favorable for the methanation of $CO_2$. The activity and selectivity were remarkably affected by iron, attributable to its small particle size, facilitated CO dissociation, and tailored surface basicity. With the best catalyst (Fe/Ni = 0.1), the $CO_2$ conversion rate was 6.96 mmol $CO_2$ $mol_{Fe+Ni}^{-1}$ $s^{-1}$ at 608 K, with a consistent selectivity of 99.3% toward $CH_4$ over 24 h on stream [46]. In contrast, Wang et al. used calcined olivine ($(Mg, Fe)_2SiO_4$) as support for Ni catalysts that were prepared by the incipient wetness method, and applied them for $CO_2$ methanation. The results showed that a $FeO_x$ phase was formed on the surface of the calcined olivine and that the unreduced $FeO_x$ between the active Ni-Fe alloy phase and the olivine support played a crucial role in $CO_2$ methanation. A 98% $CO_2$ conversion and a selectivity of 99% toward $CH_4$ were achieved at a temperature of 673 K, a $H_2/CO_2$ molar ratio of 6, and an hourly space velocity of 11,000 $h^{-1}$ [47].

Table 1 gives a list of experimental studies with Ni- and Fe-based catalysts on different support materials for $CO_2$ methanation.

**Table 1.** Experimental studies with Ni- and Fe-based catalysts on different support materials for $CO_2$ methanation.

| Catalyst Composition | Preparation Method | Operation Conditions | Performance | Ref. |
|---|---|---|---|---|
| Ni/MgO ($w_{Ni}$ = 0–27 wt%) | wet impregnation | $T$ = 533–648 K GHSV = 3.7 $m^3$ $kg^{-1}$ $h^{-1}$ $H_2$:$CO_2$:$N_2$ = 4:1:5 | $X_{CO_2}$ = 87% $S_{CH_4}$ = 99% | [9] |
| Mg-Al-$CO_3$ LDH [1] catalyst | coprecipitation | $T$ = 473–573 K $CO_2$:$O_2$:$N_2$ = 14:4:82 | $CO_2$ sorption: 2.72% (dry sorption), 3.14% (wet condition, 12% water) | [48] |
| Ni/$CeO_2$ | wet impregnation | $T$ = 573 K $F$ = 3000 mL $min^{-1}$ | $X_{CO_2}$ > 90% $S_{CH_4}$ > 99.5% | [49] |
| Ni/$Al_2O_3SiO_2$ | sol-gel | $T$ = 625 K $H_2$:$CO_2$ = 3.5:1 GHSV = 12,000 mL $g^{-1}$ $h^{-1}$ | $X_{CO_2}$ = 77% $S_{CH_4}$ = 98% | [50] |
| Ni/$ZrO_2$, Ni-K/$ZrO_2$ and Ni-La/$ZrO_2$ | wetness impregnation | $T$ = 523–723 K $H_2$:$CO_2$ = 12.5:1 $F$ = 50–100 mL $min^{-1}$ | $X_{CO_2}$ = 20–60% $S_{CH_4}$ = 89–99% | [51] |

**Table 1.** *Cont.*

| Catalyst Composition | Preparation Method | Operation Conditions | Performance | Ref. |
|---|---|---|---|---|
| Ni-FeAl-$(NH_4)_2CO_3$ | co-precipitation | $T$ = 493 K $CO_2$:$H_2$:$N_2$ = 1:4:1.7 WHSV = 9600 mL $g^{-1}$ $h^{-1}$ | $X_{CO_2}$ = 58.5% $S_{CH_4}$ = 64% | [52] |
| Ni/MCM-41 with $VO_x$-modified | | $T$ = 673 K WHSV = 60,000 mL $g^{-1}$ $h^{-1}$ | $X_{CO_2}$ = 81.4% $S_{CH_4}$ = 72.8% | [53] |
| Ni-Fe/S16 | mesoporous silica molecular sieve | $T$ = 473–573 K $H_2$:CO:$N_2$ = 3:1:1 WHSV = 15,000 mL $g^{-1}$ $h^{-1}$ | $X_{CO}$ = 100% (at 503 K) $S_{CH_4}$ > 90% | [54] |
| Ni-Fe/olivine $((Mg_xFe_{1-x})_2 SiO_4)$ | wet impregnation | $T$ = 673 K $H_2$:$CO_2$ = 6:1 GHSV = 11,000 $h^{-1}$ | $X_{CO_2}$ = 98% $S_{CH_4}$ = 99% | [47] |
| 1–10 wt% Fe/13X | wet impregnation | $T$ = 473–823 K $P$ = 1–15 bar | $X_{CO_2}$ = 74% ($T$ = 823 K, $P$ = 15 bar) $S_{CH_4}$ = 76% ($P$ = 5–15 bar) | [42] |
| 10Ni–Fe/$C_{A-C}$ [2] | wet impregnation | $T$ = 473–773 K $H_2$:$CO_2$ = 4:1 GHSV = 12,000 $h^{-1}$ | $X_{CO_2}$ = 5–74% $S_{CH_4}$ > 90% | [55] |
| $\gamma$-$Fe_2O_3$(n) and | commercial from Sigma-Aldrich | $T$ = 523–723 K $H_2$:$CO_2$ = 4:1 $F$ = 500 mL $min^{-1}$ GHSV = 120,000 $h^{-1}$ | $X_{CO_2}$ = 45–65% (65% at 723 K) $S_{CH_4}$ = 45–77% (77% at 623 K) | [56] |
| $\alpha$-$Fe_2O_3$(PVA) | polyvinyl alcohol (PVA) route | $T$ = 523–723 K $H_2$:$CO_2$ = 4:1 $F$ = 500 mL $min^{-1}$ GHSV = 120,000 $h^{-1}$ | $X_{CO_2}$ = 96% (723 K) $S_{CH_4}$ = 11% (673 K) | |

[1] layered double hydroxide, [2] C is a carbon support.

In several studies, iron-bearing materials, for instance, iron-bearing minerals such as siderite ore ($FeCO_3$), were used for catalyst preparation for a variety of applications and processes [56–59]. For instance, Hadjltaief et al. studied two natural samples, natural Tunisian hematite and siderite, as catalysts for the photocatalytic degradation of 4-chlorophenol (4-CP) in aqueous solution. Siderite exhibited higher photocatalytic oxidation activity than hematite at pH 3. Use of the siderite catalyst gave 100% conversion of 4-CP and 54% TOC removal. In terms of the removal of several organic compounds in an aqueous condition, the work confirmed that natural materials can be used as catalysts [60]. Wei et al. used siderite (doped with Mn and Ce) for the selective catalytic reduction (SCR) of $NO_x$ by $NH_3$. The siderite catalysts showed high efficiency for the removal of $NO_x$ ($NO_x$ conversions were higher than 90% at $T$ = 513–573 K and $T_{calcined}$ = 723 K). A 3% Mn/1% Ce-siderite catalyst also showed high resistance against sulfur poisoning (the $NO_x$ conversion remained above 75% after introducing 0.01% $SO_2$ in the feed for 7.5 h) [61]. Furthermore, Görmez et al. studied the use of rhombohedral $FeCO_3$ that was synthesized hydrothermally as a catalyst in the electro-fenton oxidation of *p*-benzoquinone. 95% of the total organic carbon was removed at 400 mA current. Increasing catalyst dosage had a beneficial effect on the mineralization of *p*-benzoquinone [62].

These studies show the beneficial catalytic effect of iron in various aspects and highlight its use by exploiting natural iron resources as abundant catalysts or catalyst support materials. Siderite ore, for instance, is an important source for iron and steel production in Austria [63] and China [64]. In general, siderite ore is converted to blast furnace-grade hematite through roasting in air in the sinter plant (Equation (4)).

$$FeCO_3 + 0.25\,O_2 \rightleftharpoons 0.5\,Fe_2O_3 + CO_2 \qquad (4)$$

In the context of decarbonizing iron and steel production, a novel direct reduction process (Equation (5)) for siderite ore with hydrogen was developed, which can be combined with subsequent catalytic $CO_2$ hydrogenation [65–67].

$$FeCO_3 + H_2 \rightleftharpoons Fe + H_2O + CO_2 \tag{5}$$

It was shown that during the direct reduction process, not only is $CO_2$ released from the carbonaceous ore, but also CO and methane are formed [65]. Moreover, Bock et al. suggested the application of this inexpensive and abundant natural siderite ore for energy storage with combined hydrogen and heat release [68].

The above-stated publications clearly show the beneficial effect of iron during $CO_2$ methanation and the potential of using natural iron-bearing minerals with regard to catalysis. Methane formation during the direct reduction of siderite ore suggests that this iron ore has a certain catalytic effect on $CO_2$ methanation. However, its potential as a catalyst for $CO_2$ methanation has not been evaluated yet. This raises the question of whether siderite ore can be considered a catalyst or catalyst support material for Ni-based catalysts for $CO_2$ methanation. For this purpose, Ni-based catalysts on MgO as a support material may act as benchmark catalysts in this study. The advantageous effect of MgO as a support material for nickel catalysts in $CO_2$ methanation has already been described. This in turn raises the question of whether the catalytic performance of Ni/MgO catalysts can be increased by adding siderite ore, which could create synergies and provide iron species acting as catalyst promoters.

Thus, the aim of this study was to investigate the potential of abundantly available siderite ore as a cheap raw material source for catalyst production for $CO_2$ methanation. Both its sole catalytic effect and a possible synergistic effect with MgO as a support material for Ni-based catalysts were considered. For this purpose, in a hydrogen atmosphere reduced siderite ore with and without nickel doping was used. Furthermore, the interplay of mixed reduced siderite ore/MgO support materials was examined, possibly opening a path to abundant, inexpensive bifunctional catalysts for $CO_2$ methanation.

## 2. Results and Discussion

The catalytic performance of hydrogen-reduced siderite ore for $CO_2$ hydrogenation to methane was investigated. The use of unreduced siderite ore is not reasonable, as otherwise $CO_2$ would constantly be released from the ore during the hydrogenation reaction, which would change the catalyst composition and its properties during the process.

In addition, the catalytic effect of Ni-based catalysts on support materials of MgO and siderite ore reduced with hydrogen prior to their application was evaluated. The process conditions were chosen based on the work of Sommerbauer et al. [39] and Loder et al. [9]. The molar feed gas ratio of $CO_2$:$H_2$ was 4:1. Inert nitrogen was added to the feed gas stream for balancing purposes ($H_2$:$CO_2$:$N_2$ = 56:14:30). From the literature it can be deduced that the availability of adsorbed hydrogen is a limiting factor for the rate of reaction of $CO_2$ methanation. Loder et al. investigated this effect and varied the $H_2$:$CO_2$ ratio in the feed gas from 3:1 to 5:1 [9]. As expected, the $CO_2$ conversion rose with rising hydrogen concentrations in the feed gas stream, yielding a maximum $CO_2$ conversion of 98% (equilibrium conversion: 99.8%) for the $H_2$:$CO_2$ ratio of 5:1. However, as most of the experiments in the study of Loder et al. were performed with a stoichiometric feed gas ratio of $CO_2$:$H_2$ = 4:1, for reasons of comparability, this ratio was also chosen in this study.

The performance of the different catalysts for $CO_2$ methanation was investigated at temperatures of 548 K, 598 K, and 648 K, and feed gas flow rates of 8.02, 11.32, and 14.66 $m^3 \ kg^{-1} \ h^{-1}$ (STP), referring to the flow rate of the feed gas stream per mass of catalyst.

To provide a baseline reference, the catalytic effect of the support material MgO was also tested and compared to the undoped, reduced siderite ore. Then, both—in hydrogen-reduced siderite ore and MgO—were used as support materials and were doped with Ni in various amounts (Ni loading from 22 to 31 wt%).

## 2.1. Catalytic Effect of Reduced Siderite Ore and MgO

As a baseline reference, the catalytic effect of hydrogen-reduced siderite ore was studied and compared to the catalytic performance of undoped MgO for $CO_2$ hydrogenation at 548 K, 598 K, and 648 K, respectively.

For the reduced siderite ore, siderite ore was reduced in a hydrogen atmosphere at two different reduction temperatures ($T_{red}$): 773 K and 973 K, respectively. The reduced ore was then removed from the reactor and kept at atmospheric conditions so that it was partially reoxidized and reached a stable state at atmospheric conditions. After that, the reduced ore was filled back into the reactor to perform the $CO_2$ hydrogenation experiments. The effect of the reduction temperature during siderite ore reduction was then evaluated regarding the characteristics of the support material during $CO_2$ hydrogenation.

For the undoped MgO catalyst, undoped $MgCO_3$ was prepared as described in Section 3.1, with the $MgCO_3$ being calcined in a muffle furnace with air at 723 K for 2 h and at 823 K for a further 5 h.

The $CO_2$ hydrogenation was carried out with the help of the undoped materials at different feed gas flow rates (8.02, 11.32, and 14.66 $m^3 \, kg^{-1} \, h^{-1}$) and a constant feed gas ratio of $H_2$:$CO_2$:$N_2$ = 56:14:30, and at reaction temperatures of 548 K, 598 K, and 648 K, respectively. The reaction temperatures were chosen because temperatures below 700 K are known to promote $CO_2$ methanation over CO formation from $CO_2$ [9].

### 2.1.1. Reduced Siderite Ore as a Catalyst

First, the catalytic effect of reduced siderite ore was proven for methane formation from $CO_2$. As depicted in Figure 1, the reduction temperature of siderite ore influences its performance during $CO_2$ hydrogenation, both regarding $CO_2$ conversion (a) and selectivity toward $CH_4$ (b). When the $CO_2$ hydrogenation was carried out at temperatures of 548 K and 598 K, the $CO_2$ conversion was below 5%, and the two siderite ore samples, that were reduced at different temperatures, showed no clear difference regarding their catalytic effect. At a reaction temperature of 648 K, the siderite ore sample that was reduced at $T_{red}$ = 973 K showed a pronounced catalytic effect, yielding $CO_2$ conversions of 12−15%. At all reaction temperatures, the siderite ore samples that were reduced at $T_{red}$ = 973 K gave higher $CH_4$ selectivities: 19.0% at 548 K and 5% at 598 K and 648 K, respectively. Compared with siderite ore reduction at $T_{red}$ = 773 K, the higher reduction temperature of $T_{red}$ = 973 K enhanced both the $CO_2$ conversion and the selectivity toward $CH_4$. At $T_{red}$ = 973 K, at the lowest feed gas flow rate (8.02 $m^3 \, kg^{-1} \, h^{-1}$), which meant the highest residence time in the reactor, the highest $CO_2$ conversion (19.9%) was obtained, while the highest $CH_4$ selectivity (19.0%) was obtained at the higher feed gas flow rates (11.32–14.66 $m^3 \, kg^{-1} \, h^{-1}$). Since the effect of increasing methane selectivity with increasing feed gas flow rate and thus lower residence time was only observed at the lowest reaction temperature of 548 K and only the methane selectivity at a feed gas flow rate of 8.02 $m^3 \, kg^{-1} \, h^{-1}$ differed from the one at 11.32 and 14.66 $m^3 \, kg^{-1} \, h^{-1}$, respectively, it is assumed that this value should rather be regarded as an outlier.

As the reaction temperature increased, the $CH_4$ selectivity decreased. CO was formed instead, showing that the reverse water-gas shift reaction was the dominant reaction. This is consistent with the findings of Lux et al., who investigated the direct reduction process of siderite ore with hydrogen, called reductive calcination in the study, with the aim of optimizing the process parameters to maximize the methane yield. Since it can be assumed that $CO_2$ is released from iron carbonate during direct reduction with hydrogen in a first step, and is further reduced to methane in a subsequent catalytic step, the results are directly transferable. As expected, in their study, it was proven that methane formation is favored at low temperatures and increased pressure, whereas the formation of CO is favored at high temperatures and low pressure [67].

The direct reduction of siderite ore with hydrogen has been extensively investigated for iron production. Loder et al., for instance, investigated the reaction kinetics and gave a detailed report on the degree of metallization (=mass of elemental iron per total mass of iron

in the reduced ore) for different reduction temperatures under atmospheric conditions [66]. The direct reduction experiments were carried out with the same original siderite ore from the Styrian Erzberg, with a feed gas ratio of $H_2:CO_2:N_2 = 56:14:30$ and a feed gas flow rate of 0.05 $m^3$ $h^{-1}$ (STP) (=0.0083 $m^3$ $kg^{-1}$ $min^{-1}$ (STP)) at ambient pressure. The effect of the reduction temperature was investigated in a temperature range of 773−1023 K. After the reduction experiments, the reduced ore was removed from the reactor under inert nitrogen conditions and analyzed for its degree of metallization. At a reduction temperature of 773 K, the degree of metallization of the reduced ore was 40%. At a reduction temperature of 973 K, a degree of metallization of 89% was obtained.

As the handling of catalysts under ambient conditions in air is easier and the impregnation of the reduced ore requires handling under ambient conditions and exposure to the solution anyway, the behavior of the reduced siderite ore when exposed to air was evaluated in this study. For this purpose, the same direct reduction experiments were carried out, but the removal of the reduced ore from the reactor was done under ambient conditions in air as compared to emptying the reactor and keeping the reduced siderite ore under an inert nitrogen atmosphere. For a reduction temperature of 973 K, this resulted in a partial reoxidation of the reduced iron species and thus a reduced degree of metallization of 58.2%, as compared with 89% under inert nitrogen conditions. The remaining iron fraction was split up into 32.6% $Fe^{2+}$ and 9.1% $Fe^{3+}$ for the partially reoxidized reduced siderite ore sample. This means that the majority of iron was present as metallic iron $Fe^0$ (58.2%) together with smaller fractions of $Fe^{2+}$ (32.6%) and $Fe^{3+}$ (9.1%) when siderite ore was reduced in a hydrogen atmosphere at 973 K and finally kept under ambient conditions in air. The determination of the chemical composition of the reduced (and partially oxidized) siderite ore samples is described in Section 3.4.3.

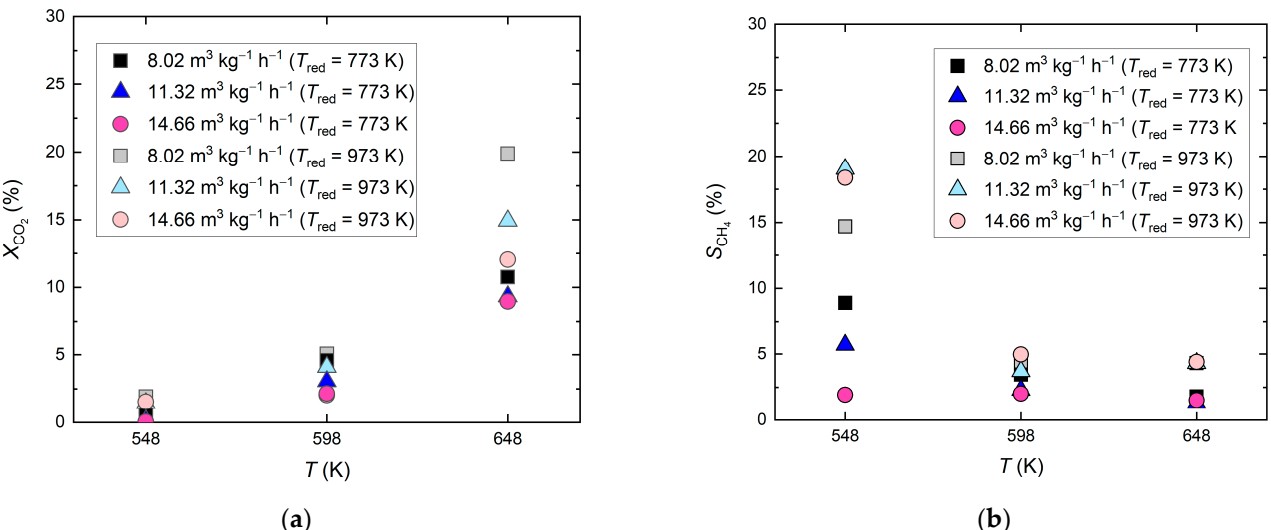

(**a**)  (**b**)

**Figure 1.** Catalytic effect of reduced siderite ore during $CO_2$ hydrogenation; reduction of siderite ore in hydrogen atmosphere at $T_{red}$ = 773 K and 973 K; molar feed gas ratio $H_2:CO_2:N_2 = 56:14:30$, feed gas flow rate 8.02–14.66 $m^3$ $kg^{-1}$ $h^{-1}$ (STP), reaction temperatures 548–648 K; (**a**) $CO_2$ conversion and (**b**) $CH_4$ selectivity.

To conclude, two different reduction temperatures ($T_{red}$ = 773 and 973 K) were studied for the preparation of the reduced siderite ore via direct reduction with hydrogen in a tubular reactor under ambient pressure. The reduction temperature was chosen according to suggestions in the literature for direct reduction of siderite ore [65,66]. Via this procedure, $CO_2$ was released from the ore, and the carbonaceous iron ore was reduced to elemental iron with minor remaining fractions of $Fe^{2+}$ and $Fe^{3+}$. When exposed to air under ambient conditions, the reduced siderite ore was partially oxidized, and for the siderite ore reduced at 973 K, more than half of the iron was still present as metallic iron $Fe^0$ (58.2%), together

with smaller fractions of $Fe^{2+}$ (32.6%) and $Fe^{3+}$ (9.1%). The degree of metallization strongly depends on the reduction temperature. Furthermore, reduction at higher temperatures results in a reduced iron ore that is more chemically stable against reoxidation. When reduced siderite ore was used as the sole catalyst for $CO_2$ hydrogenation/methanation, higher $CO_2$ conversions were obtained with the siderite ore that was reduced at the higher reduction temperature of 973 K. This may be attributed to the higher fraction of metallic iron in the reduced siderite ore. However, $CO_2$ conversions were comparably low, as the maximum $CO_2$ conversion was around 20%, and the selectivity toward methane even remained below 20%.

### 2.1.2. Undoped MgO

Second, undoped MgO was used for $CO_2$ hydrogenation in the same reaction conditions as with the reduced siderite ore. In this case, $CO_2$ was hardly converted. $CO_2$ conversions were below 0.7%, as depicted in Figure 2. Methane was not detected in the product gas.

The catalytic effect of MgO during $CO_2$ hydrogenation was already reported by Loder et al. [9], who postulated that MgO shows minor catalytic activity for the reverse water-gas shift reaction but does not promote methane formation [9]. This was confirmed by the findings in this work.

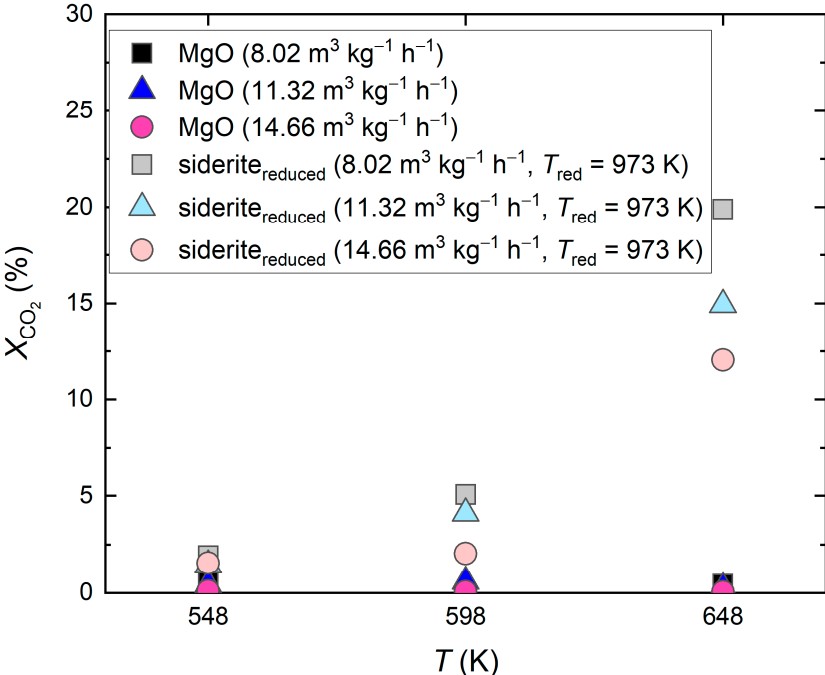

**Figure 2.** Catalytic effect of undoped MgO and reduced siderite ore (siderite$_{reduced}$) during $CO_2$ hydrogenation; reduction of siderite ore in hydrogen atmosphere at $T_{red}$ = 973 K; molar feed gas ratio $H_2$:$CO_2$:$N_2$ = 56:14:30, feed gas flow rate 8.02–14.66 m$^3$ kg$^{-1}$ h$^{-1}$ (STP), reaction temperatures 548–648 K.

### 2.2. Ni-Based Catalysts on Reduced Siderite Ore as Support Material

Next, siderite ore reduced in a hydrogen atmosphere was studied as a support material for Ni-based catalysts (Figure 3). As undoped siderite ore that was reduced at $T_{red}$ = 973 K had shown a pronounced catalytic effect during $CO_2$ hydrogenation as compared with the siderite ore that was reduced at $T_{red}$ = 773 K, a reduction temperature of $T_{red}$ of 973 K was chosen for siderite ore reduction in hydrogen prior to loading with nickel.

The effect of Ni loading was investigated at different reaction temperatures and feed gas flow rates at ambient pressure. Ni loading was varied from 22 wt% Ni (Figure 3a) to 24 wt% Ni (Figure 3b), 25 wt% Ni (Figure 3c), and 27 wt% Ni (Figure 3d) for feed gas flow

rates of 8.02, 11.32, and 14.66 m$^3$ kg$^{-1}$ h$^{-1}$, and reaction temperatures of 548 K, 598 K, and 648 K.

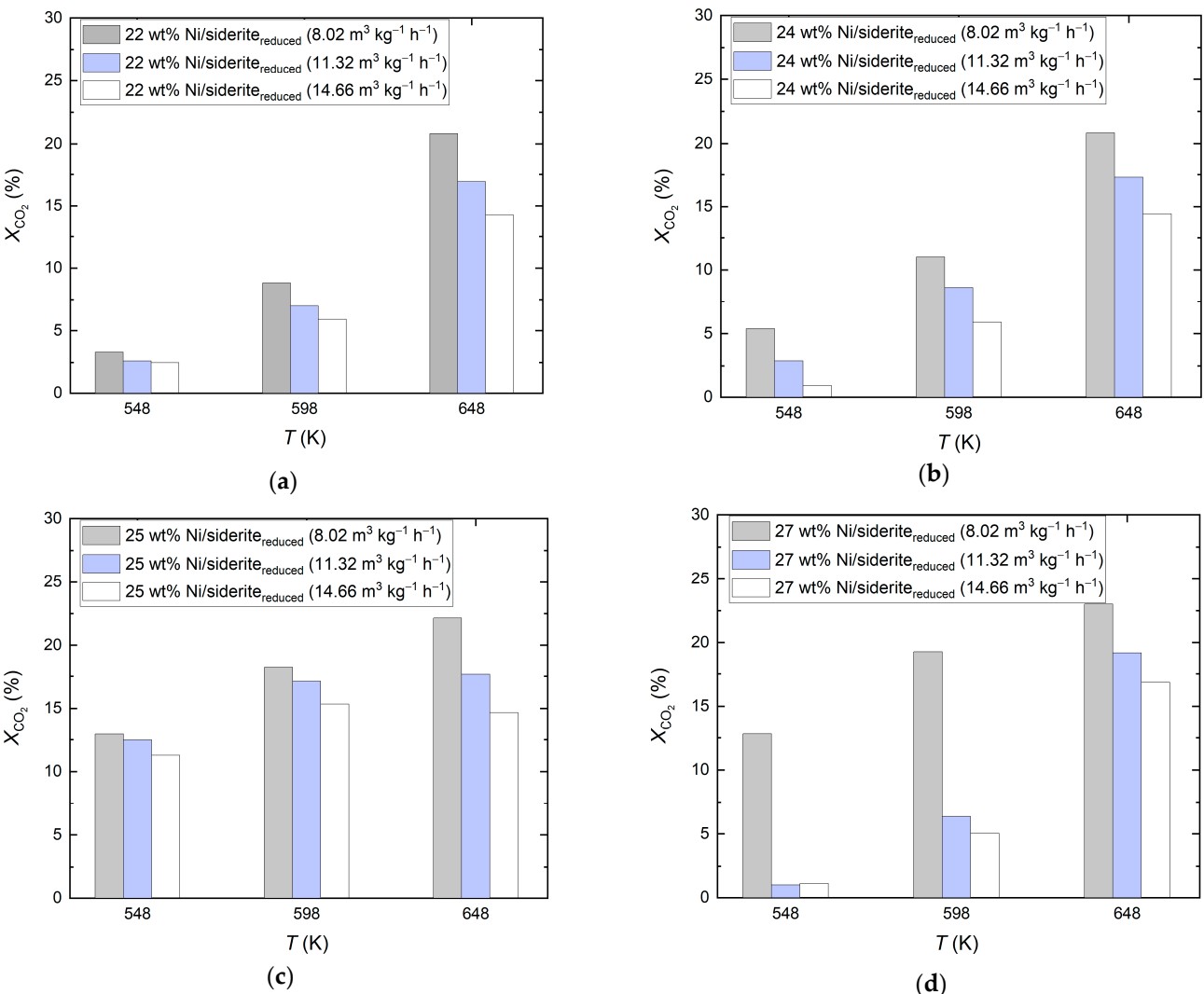

**Figure 3.** Catalytic performance of Ni-based catalytsts on reduced siderite ore as support material during CO$_2$ hydrogenation; molar feed gas ratio H$_2$:CO$_2$:N$_2$ = 56:14:30, feed gas flow rate 8.02–14.66 m$^3$ kg$^{-1}$ h$^{-1}$ (STP), reaction temperature 548 K, 598 K, and 648 K; (**a**) 22 wt% Ni/siderite$_{reduced}$, (**b**) 24 wt% Ni/siderite$_{reduced}$, (**c**) 25 wt% Ni/siderite$_{reduced}$, and (**d**) 27 wt% Ni/siderite$_{reduced}$.

For all Ni loadings of the Ni/siderite$_{reduced}$ catalysts, the CO$_2$ conversion increased with increasing reaction temperature and decreasing feed gas flow rate (increasing residence time). The highest CO$_2$ conversion was 23.8% and was obtained with the 27 wt% Ni/siderite$_{reduced}$ catalyst at a reaction temperature of 648 K. The effect of Ni loading on the CO$_2$ conversion and the CH$_4$ selectivity at a constant feed gas flow rate of 8.02 m$^3$ kg$^{-1}$ h$^{-1}$ is depicted in Figure 4. Both the CO$_2$ conversion and the CH$_4$ selectivity increased with increasing Ni loading. At reaction temperatures of 548 K and 598 K, the CO$_2$ conversion increased significantly with increasing Ni loading from 22 wt%/24 wt% to 25 wt%, but only slightly increased further for a Ni loading of 27 wt%. At a reaction temperature of 648 K, only a minor effect of Ni loading on the CO$_2$ conversion was visible. As opposed to the findings with undoped support material, with increasing reaction temperatures, the selectivity toward CH$_4$ increased.

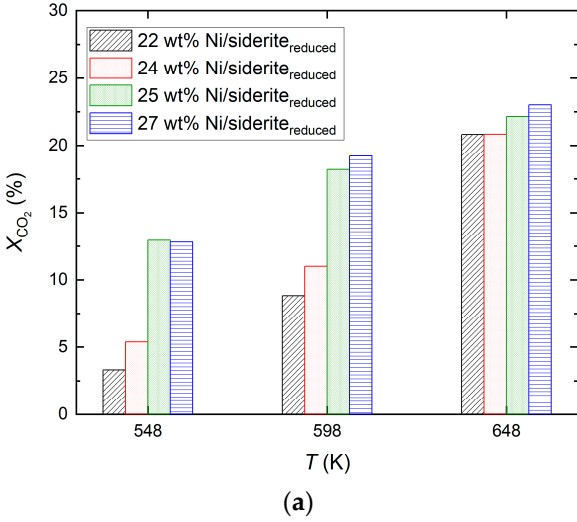
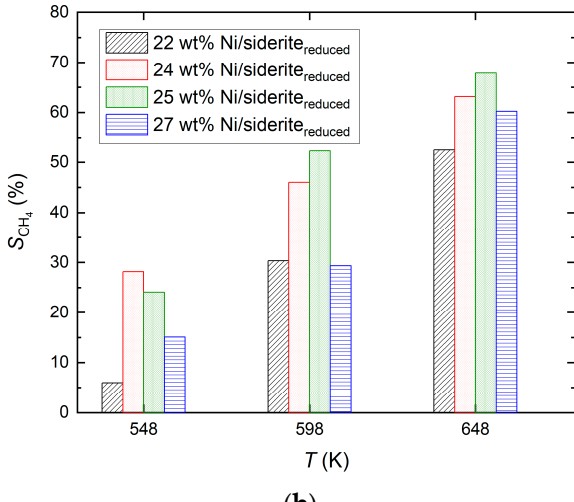

**Figure 4.** Effect of Ni loading of Ni/siderite$_{reduced}$ catalysts on $CO_2$ conversion (**a**) and selectivity toward $CH_4$ (**b**) during $CO_2$ hydrogenation; molar feed gas ratio $H_2$:$CO_2$:$N_2$ = 56:14:30, feed gas flow rate 8.02 m$^3$ kg$^{-1}$ h$^{-1}$ (STP), reaction temperatures 548 K, 598 K, and 648 K.

The catalytic activity of the 27 wt% Ni/siderite$_{reduced}$ catalyst adopts an exceptional position in the range of Ni/siderite$_{reduced}$ catalysts. Its catalytic activity in terms of $CO_2$ conversion is only marginally higher than that of the 25 wt% Ni/siderite$_{reduced}$ catalyst. At all three reaction temperatures, however, it shows a significantly reduced selectivity for methane. Furthermore, the 27 wt% Ni/siderite$_{reduced}$ catalyst shows poor catalytic activity when the feed gas flow rate is high and, thus, the residence time is low. This suggests that in this range, a maximum Ni load is present, which is advantageous in terms of methane selectivity. Furthermore, it must be noted that the production of catalysts with such a high Ni loading on reduced siderite ore is more difficult to reproduce, which could also be reflected in the experimental results. However, at low feed gas flow rates (Figure 4), the catalyst with the highest Ni loading (27 wt% Ni/siderite$_{reduced}$) gave the highest $CO_2$ conversion (23.0%) and high $CH_4$ selectivity (60.2%). Thus, a high Ni loading (26–28 wt% Ni) was chosen in the subsequent study to investigate the catalytic performance of Ni-based catalysts on mixed reduced siderite ore/MgO support material.

### 2.3. Mixed Reduced Siderite Ore/Magnesium Oxide as Support Material for Ni-Based Catalysts

In previous studies, MgO has already been proven to be an effective support material for Ni-based catalysts, giving access to bifunctional Ni/MgO catalysts [9,39]. In this study, a 31 wt% Ni/MgO catalyst was prepared and used as a benchmark catalyst.

In order to test the catalytic performance of mixed MgO and reduced siderite ore as support materials for Ni-based catalysts, various ratios of reduced siderite ore and MgO were tested at constant Ni loading of 28 wt%; siderite$_{reduced}$/MgO (*w/w*) = 30:70, 50:50, and 70:30, respectively.

As depicted in Figure 5, Ni/siderite$_{reduced}$/MgO catalysts efficiently catalyze $CO_2$ hydrogenation. As expected, for all three catalysts—siderite$_{reduced}$/MgO = 30:70, 50:50, 70:30—the $CO_2$ conversion increases with increasing reaction temperature and decreasing feed gas flow rate. The highest $CO_2$ conversions were obtained with the catalysts with a higher fraction of reduced siderite (siderite$_{reduced}$/MgO = 50:50, 70:30), with $CO_2$ conversions of 74.2−78.0%, 67.0−69.95%, and 63.5−65.8% at a reaction temperature of 648 K and feed gas flow rates of 8.02, 11.32, and 14.66 m$^3$ kg$^{-1}$ h$^{-1}$, respectively.

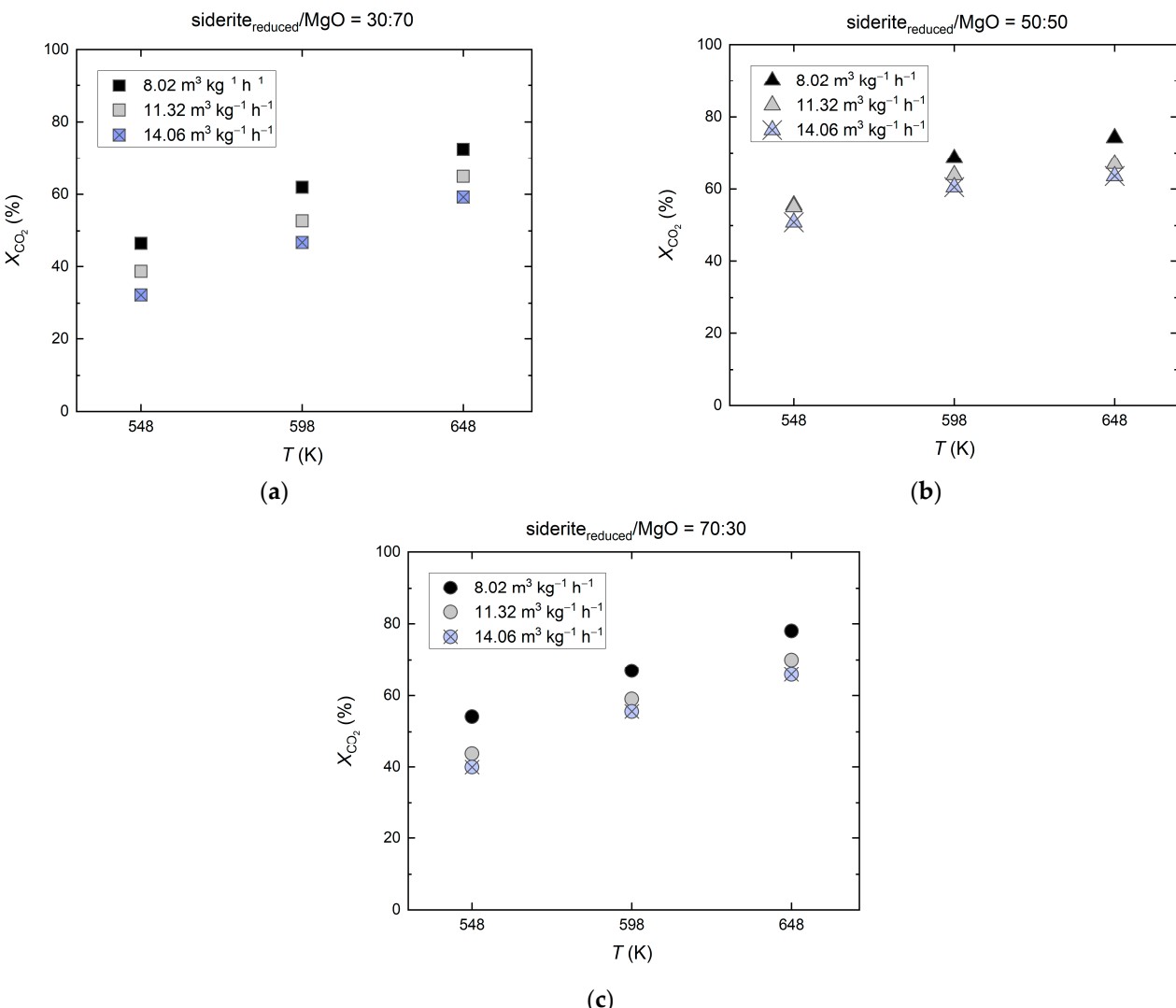

**Figure 5.** Catalytic performance of Ni-based catalysts (28 wt% Ni) on mixed $siderite_{reduced}$/MgO support material in various compositions for $CO_2$ hydrogenation; molar feed gas ratio $H_2$:$CO_2$:$N_2$ = 56:14:30, feed gas flow rate 8.02−14.66 m³ kg⁻¹ h⁻¹ (STP), reaction temperatures 548–648 K, reduction of siderite ore in hydrogen at $T_{red}$ = 973 K; (**a**) $siderite_{reduced}$/MgO = 30:70, (**b**) $siderite_{reduced}$/MgO = 50:50, and (**c**) $siderite_{reduced}$/MgO = 30:70.

Figure 6 depicts the catalytic performance of the 28 wt% Ni/$siderite_{reduced}$/MgO catalysts compared with Ni-based catalysts on MgO (31 wt% Ni/MgO) or reduced siderite ore (28 wt% Ni/$siderite_{reduced}$) only. It is evident that the Ni/$siderite_{reduced}$ catalyst performed the worst. When MgO was added to the support material, there was a clear improvement in catalytic performance. At lower reaction temperatures (548 K), the catalytic performance seemed to be worse with an excess of MgO ($siderite_{reduced}$/MgO = 30:70) in the support material compared with the catalysts with lower MgO fractions ($siderite_{reduced}$/MgO = 50:50, 70:30). However, the difference was small and seemed to be canceled out at higher temperatures (648 K). No difference was visible for the Ni/$siderite_{reduced}$/MgO catalysts with reduced siderite ore to MgO mass ratios of 50:50 and 70:30. $CH_4$ selectivities were >95% with all Ni/$siderite_{reduced}$/MgO catalysts.

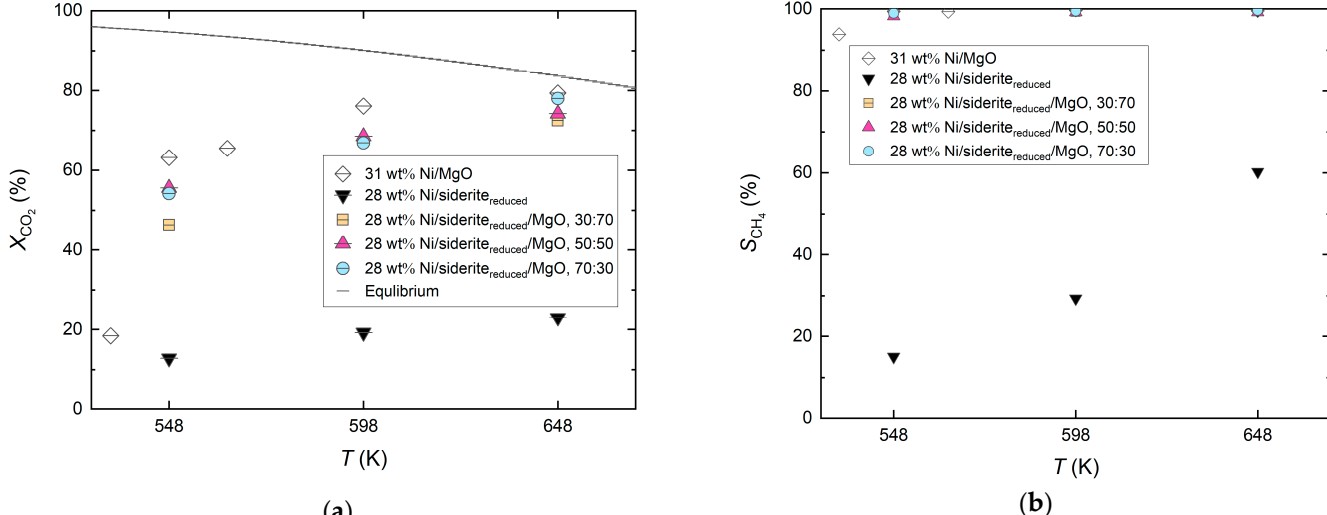

**Figure 6.** Comparison of the catalytic performance of Ni-based catalysts on reduced siderite ore, MgO, and mixed reduced siderite ore/MgO support material (27–30 wt% Ni) for various reaction temperatures (548–648 K); molar feed gas ratio $H_2:CO_2:N_2$ = 56:14:30, feed gas flow rate 8.02 $m^3$ $kg^{-1}$ $h^{-1}$ (STP), reduction of siderite ore in hydrogen at $T_{red}$ = 973 K; (**a**) $CO_2$ conversion, and (**b**) $CH_4$ selectivity.

In this series, the Ni/MgO catalyst turned out to be the best catalyst. However, it must be noted that this catalyst had a higher Ni loading of 31 wt% as compared with the 28 wt% of the Ni/siderite$_{reduced}$/MgO catalysts. This may be dedicated to the production of the Ni-doped catalysts. The doping of MgO according to the procedure described in Section 3.1 is easier than the doping of (mixed) reduced siderite ore. As a result, a higher loading was achieved, which could not be achieved at all with the mixed catalysts.

To conclude, the study revealed that reduced siderite ore acts as an efficient support material for a Ni-based catalyst for $CO_2$ methanation when combined with MgO. As expected, the $CO_2$ conversion increased with increasing Ni loading of the respective catalyst, both for Ni/siderite$_{reduced}$ and Ni/siderite$_{reduced}$/MgO catalysts, as well as with increasing temperature from 548 K to 648 K, and decreasing feed gas flow rate. When reduced siderite ore was used as sole support material (Ni/siderite$_{reduced}$ catalysts), the catalytic performance ($X_{CO2, 648 K}$ = 23.0% for 27 wt% Ni/siderite$_{reduced}$) was only marginally higher than the catalytic performance of undoped reduced siderite ore ($X_{CO2, 648 K}$ = 19.9%). This leads to the conclusion that both the iron species in and the nickel on the reduced siderite ore show a comparable catalytic effect toward $CO_2$ hydrogenation. However, the selectivity toward $CH_4$ is significantly higher when the reduced siderite ore is doped with Ni, specifically, 60.2% compared with 19.0% for undoped reduced siderite ore. This shows the high catalytic activity of the nickel species for $CH_4$ formation.

With XRD, a $NiFe_2^{3+}O_4$ (trevorite) peak was identified in all Ni/siderite$_{reduced}$ catalysts, as well as the Ni-Fe alloy and FeNi (tetrataenite) in the catalysts after $CO_2$ hydrogenation (see Section 2.4.2).

However, adding MgO to the support material of the Ni-based catalysts significantly enhanced $CO_2$ methanation. The ratio of reduced siderite ore and MgO seemed to play a subordinate role, with higher proportions of reduced siderite ore causing slightly higher $CO_2$ conversions. With ≥50% reduced siderite ore in the mixed siderite$_{reduced}$/MgO support material, no difference in the catalytic performance was visible anymore. Most importantly, adding MgO to the reduced siderite ore support drastically enhanced the selectivity toward $CH_4$. Even if only 30% of MgO was present, the selectivity toward methane approached 100%. It can be concluded that with identical Ni loading, Ni/siderite$_{reduced}$/MgO (≥30% MgO) and Ni/MgO catalysts show comparable catalytic performance, with $CO_2$ conversions close to the thermodynamic equilibrium and high $CH_4$ selectivity ≥ 99.9%.

This supports the findings of Loder et al., highlighting the fundamental role of MgO as a basic support material for Ni-based catalysts for $CO_2$ methanation [9].

## 2.4. Catalyst Characterization

### 2.4.1. X-ray Fluorescence Spectrometry

The freshly prepared 24 wt% Ni/siderite$_{reduced}$ catalyst (after calcination, before reduction of NiO to Ni with hydrogen) was analyzed by X-ray fluorescence (XRF) spectrometry for its Ni loading. The Ni loading was compared with the value obtained by AAS analysis. The XRF result showed that NiO was present, and the percentage of Ni (24.81%) was in good agreement with the AAS analysis (24.1%). The other constituents were present at 55.0 wt% $Fe_2O_3$, 7.45 wt% $SiO_2$, 2.55 wt% MgO, 1.79 wt% MnO, and 2.31 wt% CaO. This shows that during the wet preparation/impregnation procedure, elemental iron and $Fe^{2+}$ were oxidized to $Fe^{3+}$.

### 2.4.2. X-ray Diffraction

Fresh and used Ni/siderite$_{reduced}$ catalysts were analyzed by X-ray diffraction (XRD). The Ni-based catalysts on reduced siderite ore support material that were analyzed were (i) 22 wt% Ni/siderite$_{reduced}$ (fresh catalyst), (ii) 22 wt% Ni/siderite$_{reduced}$ (used catalyst), (iii) 24 wt% Ni/siderite$_{reduced}$ (fresh catalyst), (iv) 24 wt% Ni/siderite$_{reduced}$ (used catalyst), (v) 25 wt% Ni/siderite$_{reduced}$ (fresh catalyst), and (vi) 25 wt% Ni/siderite$_{reduced}$ (used catalyst), as shown in Figure 7.

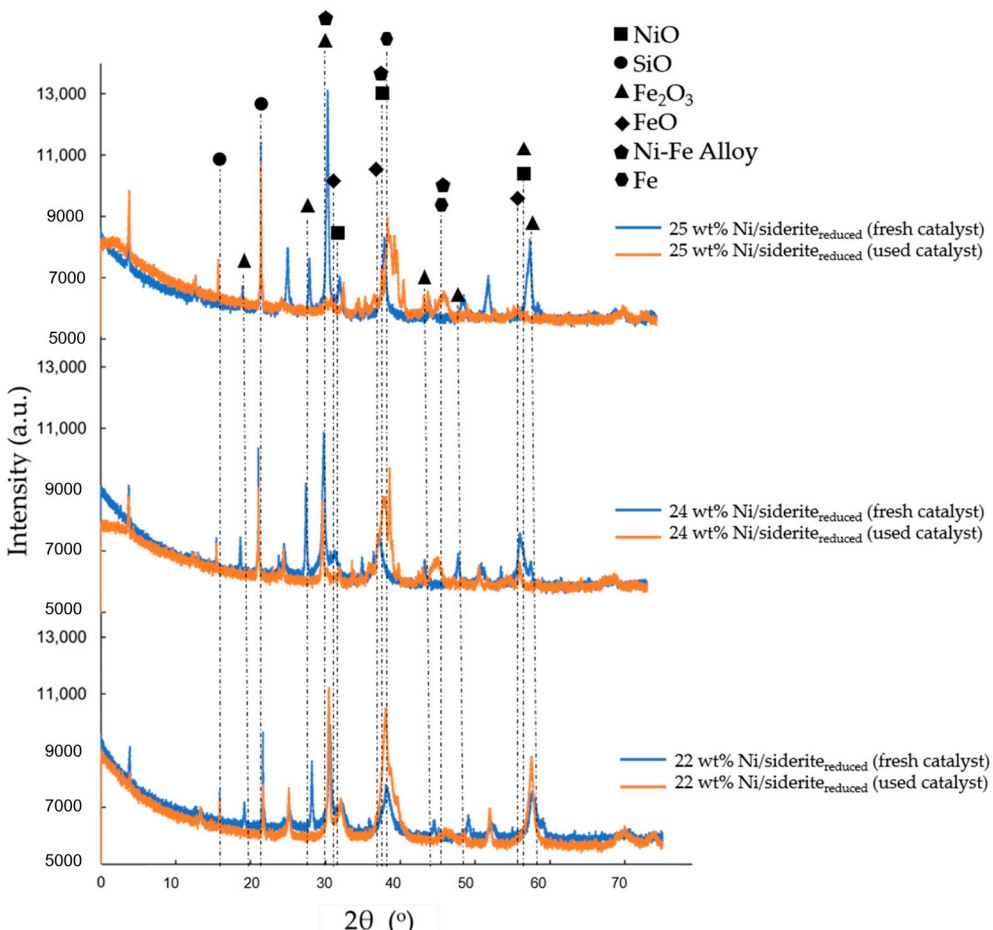

**Figure 7.** X-ray diffraction results of the fresh and used Ni/siderite$_{reduced}$ catalysts; 22 wt% Ni/siderite$_{reduced}$, 24 wt% Ni/siderite$_{reduced}$, and 25 wt% Ni/siderite$_{reduced}$.

In the fresh Ni/siderite$_{reduced}$ catalysts with 22 wt%, 24 wt%, and 25 wt% Ni loading, the diffraction peaks of NiO (bunsenite), NiFe$_2$$^{3+}$O$_4$ (trevorite), Fe$_2$O$_3$ (hematite), KAl$_2$(Si$_3$Al)O$_{10}$ (OH$_2$) (muscovite), SiO$_2$ (quartz), and FeO (wüstite) were identified. When used for CO$_2$ hydrogenation, the catalysts changed phases; NiFe$_2$$^{3+}$O$_4$ (trevorite) and Fe$_2$O$_3$ (hematite) did not appear anymore; nonetheless, Fe$^{2+}$(Fe$^{3+}$)$_2$O$_4$ (magnetite), Ni-Fe alloy, and FeNi (tetrataenite) were found in all of the used catalysts (after CO$_2$ hydrogenation with a molar feed gas ratio H$_2$:CO$_2$:N$_2$ = 56:14:30, feed gas flow rates of 8.02–14.66 m$^3$ kg$^{-1}$ h$^{-1}$ (STP), and reaction temperatures of 548 K, 598 K, and 648 K). The result showed that the Ni-Fe alloys were small crystallites with increasing % of Ni. In addition, another iron alloy was found in the 24 wt% Ni/siderite$_{reduced}$ and 25% wt% Ni/siderite$_{reduced}$ catalysts; (Fe,Ni,Co)$_3$C (cohenite) was present in the catalysts with increased Ni loading. However, the presence of significant quantities of cobalt can be excluded.

### 2.4.3. Scanning Electron Microscopy (SEM) Analysis

SEM images were taken by Field Emission Scanning Electron Microscopy from the catalyst samples: 21.69 wt% Ni/siderite$_{reduced}$ (Figure 8a), 23.77 wt% Ni/siderite$_{reduced}$ (Figure 8b), 24.80 wt% Ni/siderite$_{reduced}$ (Figure 8c), and 27.71 wt% Ni/siderite$_{reduced}$/MgO (30:70) (Figure 9a), 28.01 wt% Ni/siderite$_{reduced}$/MgO (Figure 9b) (50:50), and (c) 28.18 wt% Ni/siderite$_{reduced}$/MgO (Figure 9c) (70:30).

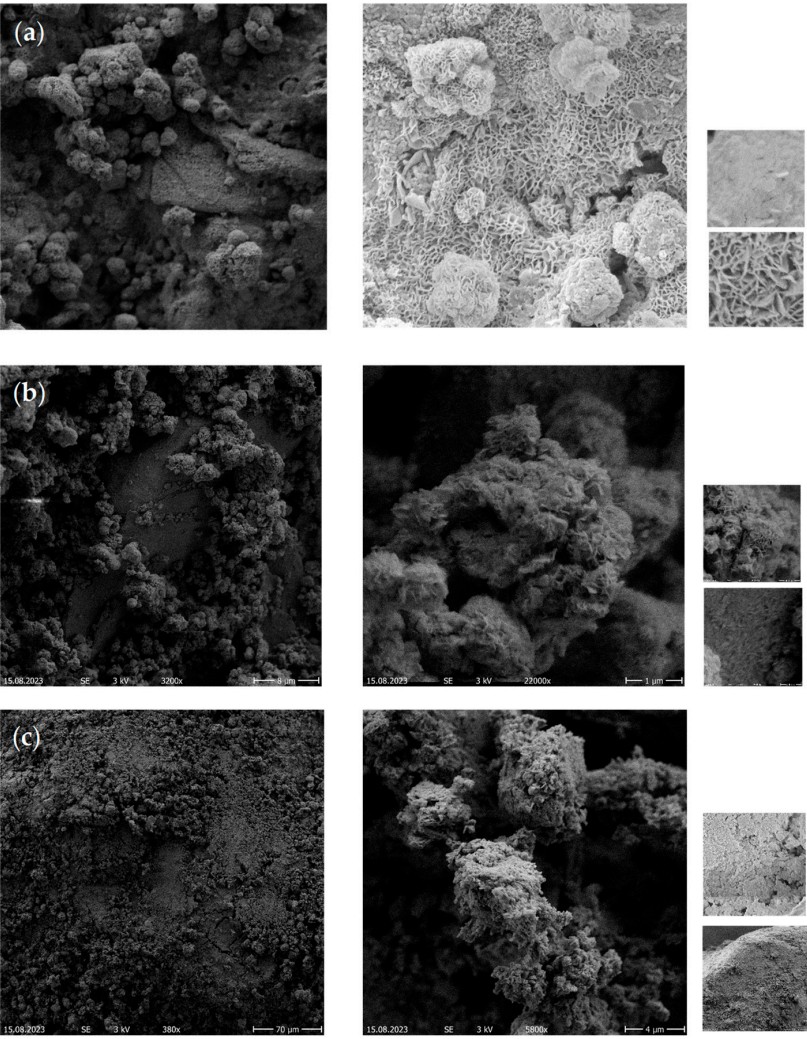

**Figure 8.** Scanning Electron Microscopy (SEM) analysis of different Ni loadings on the reduced siderite ore support material: (**a**) 21.69 wt% Ni/siderite$_{reduced}$, (**b**) 23.77 wt% Ni/siderite$_{reduced}$, and (**c**) 24.80 wt% Ni/siderite$_{reduced}$; fresh catalysts.

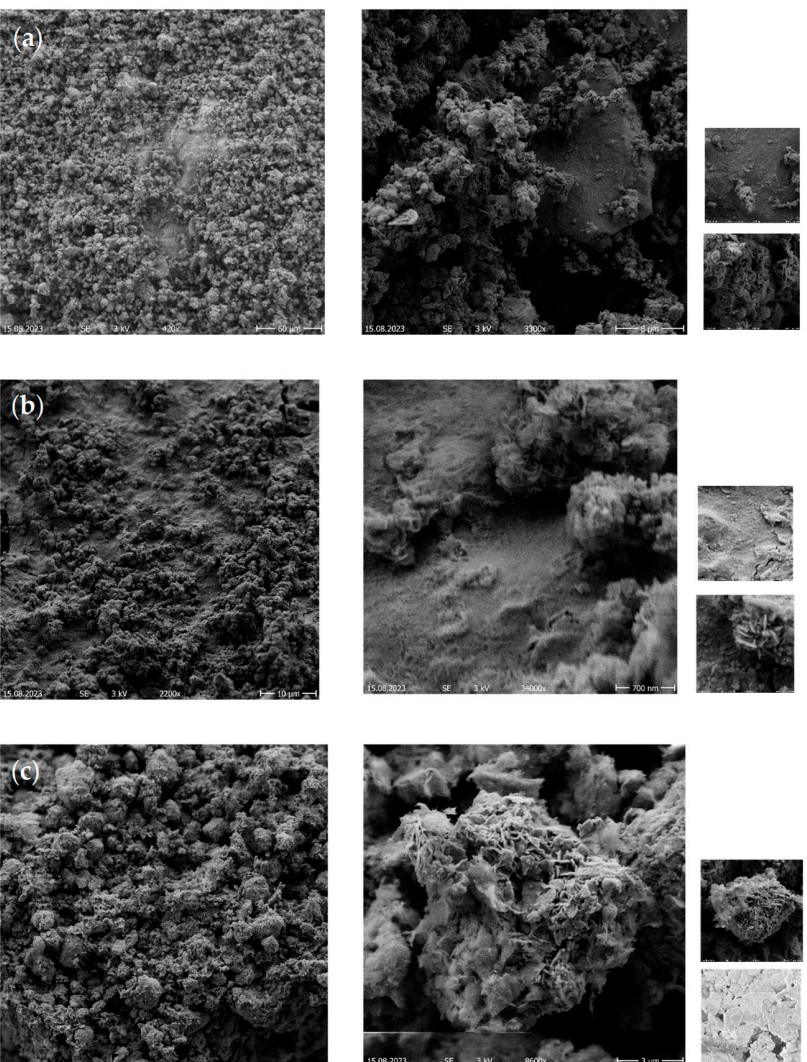

**Figure 9.** Scanning Electron Microscopy (SEM) analysis of different proportions of reduced siderite ore and MgO as support material for Ni-based catalysts: (**a**) 27.71 wt% Ni/siderite$_{reduced}$/MgO (30:70), (**b**) 28.01 wt% Ni/siderite$_{reduced}$/MgO, and (**c**) 28.18 wt% Ni/siderite$_{reduced}$/MgO.

The SEM images in Figure 8 show that the Ni/siderite$_{reduced}$ catalysts became slick and homogeneous when the Ni loading increased. For all Ni loadings, the particles had a clear distribution of NiO and FeO on the surface. At 21.69 wt% Ni loading, the catalyst had a more porous structure when compared with 23.77 wt% and 24.80 wt% Ni loading on reduced siderite ore. Thus, it can be said that surface morphology changed with increasing Ni loading.

Furthermore, adding MgO to the reduced siderite ore support (siderite$_{reduced}$:MgO = 30:70, 50:50, 70:30) resulted in more irregularities when increasing the MgO ratio (Figure 9). At siderite$_{reduced}$:MgO ratios of 30:70 and 50:50, the surface had a flat and homogeneous structure, while the catalyst with the siderite$_{reduced}$:MgO = 70:30 support had irregular particles on the surface.

## 3. Materials and Methods

### 3.1. Materials and Catalyst Preparation

Ni-based catalysts on two different support materials and combinations of the support materials were tested: (i) Ni/MgO, (ii) Ni/siderite$_{reduced}$, and (iii) Ni/siderite$_{reduced}$/MgO. The catalysts were prepared by wet impregnation with nickel nitrate hexahydrate (Ni(NO$_3$)$_2$ ·6 H$_2$O, 99%, p.a., Lactan). The MgO support was prepared from MagGran© (4MgCO$_3$·Mg

(OH)$_2$·4H$_2$O), while the siderite ore originated from the Styrian Erzberg, Austria, and was provided by VA Erzberg GmbH, Eisenerz, Austria (particle size 0.5−1 mm). Its mineral composition is given in Table 2.

**Table 2.** Chemical composition of the original siderite ore determined by XRF spectroscopy [65].

| Component | wt% |
|---|---|
| Fe | 33.56 |
| CaO | 6.88 |
| MgO | 3.72 |
| Mn | 1.91 |
| SiO$_2$ | 5.19 |
| Al$_2$O$_3$ | 1.02 |
| others (including CO$_2$) | 47.72 |

This siderite ore consists of the iron-bearing minerals siderite ((Fe$_{0.83}$Mg$_{0.11}$Mn$_{0.05}$Ca$_{0.01}$)CO$_3$) with substitutions of magnesium carbonate (MgCO$_3$), calcium carbonate (CaCO$_3$) and manganese carbonate (MnCO$_3$), and ankerite ((Ca$_{0.51}$Fe$_{0.31}$Mg$_{0.15}$Mn$_{0.03}$)CO$_3$). Furthermore, dolomite ((Ca,Mg)(CO$_3$)$_2$), calcite (CaCO$_3$), quartz (SiO$_2$), and muscovite (KAl$_2$(Al-Si$_3$O$_{10}$)(OH)$_2$) are present in this carbonaceous ore (Table 3).

**Table 3.** Composition of the original siderite ore from the Austrian Erzberg (particle size 0.5−1 mm) [65].

| Component | wt% |
|---|---|
| Siderite | 79.04 |
| Calcite | 8.91 |
| Quartz | 5.19 |
| Ankerite | 3.96 |
| Dolomite | 1.80 |
| others (including CO$_2$) | 47.72 |

The preparation of the catalysts was based on the work of Loder et al. [9] and extended for reduced siderite ore as further catalyst and catalytic support material. There are four main preparation steps:

(i)    calcination of magnesium carbonate for magnesium oxide preparation (Equation (6)) and/or reduction of siderite ore (Equation (7)) [69,70],

$$4\,MgCO_3 \cdot Mg(OH)_2 \cdot 4\,H_2O \rightarrow 5\,MgO + 4\,CO_2 + 5\,H_2O \tag{6}$$

$$FeCO_3 + (x + y + 4z)\,H_2 \rightarrow FeO_{1-x} + (1 - y - z)CO_2 + yCO + zCH_4 + (x + y + 2z)H_2O \tag{7}$$

(ii)   impregnation of the support material with nickel nitrate,
(iii)  thermal decomposition of nickel nitrate to nickel oxide (Equations (8)–(10)) [71], and

$$Ni(NO_3)_2 \cdot 6\,H_2O \rightleftharpoons NiO + 2\,NO_2 + 0.5\,O_2 + 6\,H_2O \tag{8}$$

$$MgO + Ni(NO_3)_2 \rightarrow (Mg_{1-x}Ni_x)(OH)_2 \tag{9}$$

$$(Mg_{1-x}Ni_x)(OH)_2 \rightarrow Mg_{1-x}Ni_xO + H_2O \tag{10}$$

(iv)  reduction of Ni with hydrogen (Equations (11) and (12)) [39,71].

$$Mg_{1-x}Ni_xO_2 + H_2 \rightarrow [Ni]_x\,Mg_{1-x}O + H_2O \tag{11}$$

$$NiO + H_2 \rightarrow Ni + H_2O \tag{12}$$

The experimental procedure was as follows:

(i)     Preparation of the support material:

For magnesium oxide preparation, magnesium carbonate powder ($4\,MgCO_3 \cdot Mg(OH)_2 \cdot 4H_2O$) was calcined in a muffle furnace (Heraeus M 110) with air at 723 K for 2 h and at 823 K for 5 h.

For preparation of the reduced siderite ore support (for details see Section 3.2), siderite ore was reduced in the tubular reactor used for the methanation experiments, in 90% hydrogen (feed gas ratio of $H_2:N_2$ = 9:1, at a feed gas flow rate of $0.048\,m^3\,h^{-1}$), and at 773 K and 973 K until the exit gas composition equaled the feed gas composition. The reduced siderite ore was then exposed to air at room temperature, where it partially oxidized.

(ii)    Impregnation:

The nickel nitrate solution was prepared from $Ni(NO_3)_2 \cdot 6\,H_2O$ that was mixed with ultrapure water at a nickel concentration of $53-60\,g\,dm^{-3}$ in a flask. After that, the $70\,cm^3$ nickel solution in the continuously stirred flask was cooled in a water bath ($T = 293-298$ K). 10 g of calcined MgO and/or reduced (and partially oxidized) siderite ore was added to the nickel solution (adding 1 g per 3 min). The mixed solution (slurry phase) was constantly stirred for 2 h and filtrated with the help of a vacuum pump (separated slurry phase and residual water phase). The filtrated slurry (green) phase was dried overnight at room temperature.

(iii)   Thermal deposition:

The freshly prepared and pre-dried catalysts were then dried in a muffle furnace (Heraeus M 110) at 393 K for 2 h and at 673 K for 5 h in air. After the drying process, the catalysts had a light gray color.

(iv)   Reduction with hydrogen/activation:

This step was required for the reduction of NiO to Ni. A total of 4 g of the catalyst powder was reduced with hydrogen (feed gas ratio of $H_2:N_2$ = 9:1, feed gas flow rate of $0.048\,m^3\,h^{-1}$) in the tubular reactor at 773 or 973 K (temperature measurement at $T_3$ thermocouple position, as described in Section 3.4.1, for 4 h.

Then, the catalysts were kept in the tubular reactor and used for the $CO_2$ methanation experiments.

### 3.2. Siderite Ore Reduction

The original siderite ore was reduced in a hydrogen atmosphere at two different reduction temperatures, 773 K and 973 K, at ambient pressure and a feed gas ratio of $H_2:N_2$ = 90:10. This process step is known as the direct reduction of siderite ore with hydrogen. In the literature, it is suggested for the production of elemental iron from siderite ore in a single process step [65]. The course of the direct reduction process is shown in Figure 10 via the product gas composition at the reactor exit and the reduction temperature at thermocouple position $T_3$.

CO and $CO_2$ formation started in a temperature range of $656-683$ K (heat-up phase) for both experiments (final reduction temperatures of 773 K and 973 K, respectively). At 773 K, the highest CO concentration in the product gas was 10.8%, and the highest $CO_2$ concentration was 37.7% (at the lowest $H_2$ concentration, 41.9%). At a reduction temperature of 973 K, the highest CO concentration was 13.8%, and the highest $CO_2$ concentration was found to be 46.4% at the lowest $H_2$ concentration of 35.3%. In addition, when kept under an inert nitrogen atmosphere, the degree of metallization was 40% and 89% for the reduction temperatures of 773 K and 973 K, respectively.

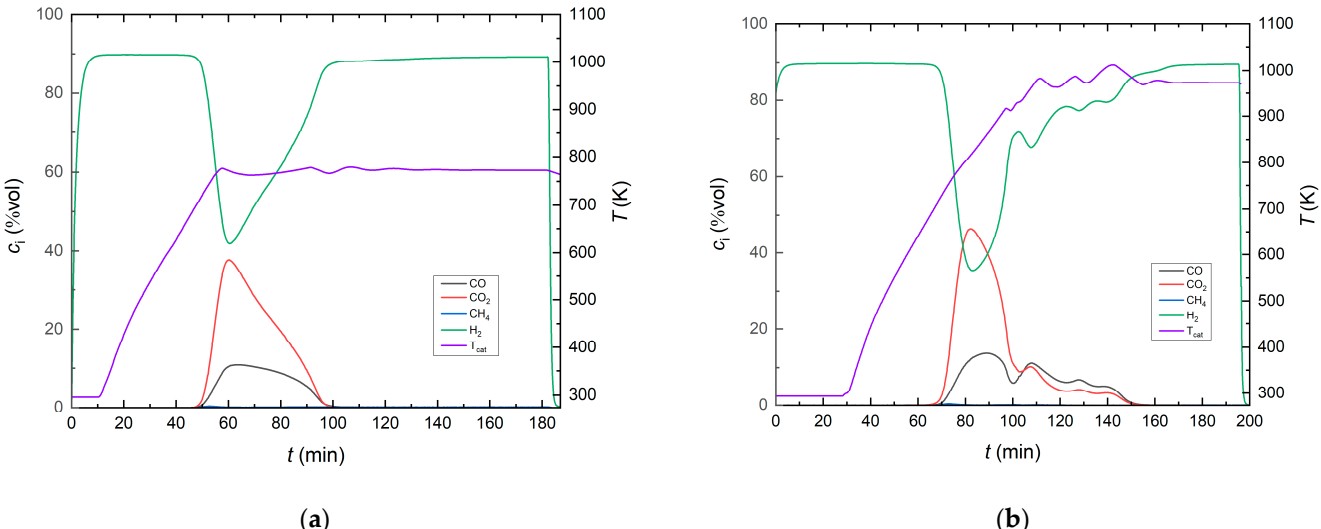

**(a)**                                                                 **(b)**

**Figure 10.** Product gas composition during direct reduction of siderite ore with hydrogen (feed gas ratio $H_2$:$N_2$ = 90:10, feed gas flow rate 0.048 $m^3\ h^{-1}$) for preparation of the support material; (**a**) $T_{red}$ = 773 K and (**b**) $T_{red}$ = 973 K.

The concomitant carbonate species in the siderite ore (magnesium, manganese, and calcium carbonate) were converted to their respective bivalent oxides, as shown in Equation (13).

$$MeCO_3 \rightleftharpoons MeO + CO_2;\ Me = Mg,\ Mn\ or\ Ca \tag{13}$$

### 3.3. Catalyst Characterization

#### 3.3.1. X-ray Diffraction

The XRD patterns of the catalyst samples were obtained by a Rigaku SmartLab® X-ray diffractometer, Rigaku Corporation, Tokyo, Japan. The samples were collected using a sweep speed of 2.0° $min^{-1}$ and 2θ = 5.0–80.0° with a scan step-size of 0.01°.

#### 3.3.2. X-ray Fluorescence Spectrometry

The catalyst samples were dried in a furnace at 378 K for 2 h to determine the loss on ignition (LOI). After that, the samples were mixed with lithium tetraborat ($Li_2B_4O_7$) and lithium metaborat ($LiBO_2$) and melted in a melting furnace at 1273 K for 1 h. An S8 TIGER Series 2 XRF wavelength dispersive (WDX) spectrometer from Bruker AXS GmbH, Karlsruhe, Germany, was used to determine the catalyst composition using the Best Detection-Vac34mm measurement method.

#### 3.3.3. Scanning Electron Microscopy

The DSM 982 Gemini Field Emission Scanning Electron Microscope (FE-SEM) from Carl Zeiss Microscopy Deutschland GmbH, Oberkochen, Germany, was used to study the surface of the Ni/siderite$_{reduced}$ and Ni/siderite$_{reduced}$/MgO catalysts. The specimen stage was from −15° to +90°, and the working resolution was between 1 and 4 nm. The samples were coated by a single Leica EM ACE600 sputter coater, Leica Mikrosysteme GmbH, Austria.

#### 3.3.4. AAS Analysis

The nickel loading of the catalysts was primarily analyzed by atomic absorption spectrometry (AAS). During the catalyst preparation (impregnation with nickel nitrate), process samples of the nickel solution (before and after impregnation of the support material) were taken and mixed with a solution of 10 $cm^3$ $HNO_3$ (Carl Roth GmbH, Karlsruhe, Germany) and 990 $cm^3$ deionized water. An AAnalyst 400 atomic absorption spectrometer (Perkin Elmer Instruments LLC, Shelton, United States of America) was used,

equipped with a nickel hollow cathode lamp set to 25 mA current at a wavelength of 232 nm, and applying a compressed air/ethylene flame, to determine the respective nickel concentrations. From the AAS results, the amount of nickel loading was determined.

### 3.4. Experimental Setup and Experimental Procedure of the Methanation Experiments

Hydrogen (99.999%), carbon dioxide (99.998%), and nitrogen (99.999%) supplied by Air Liquide were used for the $CO_2$ hydrogenation experiments. Nitrogen was used as an inert gas for heat transport and balancing purposes.

### 3.4.1. Experimental Setup

The experimental setup is shown in Figure 11. The feed gases ($H_2$, $CO_2$, and $N_2$) were controlled by three mass flow controllers (MFC1-3). They passed the fixed-bed stainless steel tubular reactor (Parr Instrument GmbH, Illinois, United States of America), which was 0.82 m in length and had an inner diameter of 25 mm. The feed gas stream was preheated by a pre-heating coil (PHC) at the top of the reactor tube before entering the catalyst bed. The tubular reactor was heated by an electric furnace with three heating zones ($HT_1$–$HT_3$) at the outer wall of the reactor tube. The temperature in the reactor was measured by six thermocouples (T1–T6). They were placed at two positions per heating zone. Stainless steel spacers and a sieve were placed inside the reactor to maintain the catalyst's position. A total of 4 g of the Ni-based catalysts was placed in the middle of the reactor with the thermocouple string (thermocouple diameter = 6 mm) at the lower end of the catalyst bed. When the gas stream had passed the catalyst bed, the gaseous product stream was cooled (i) in the cooler at the outlet of the reactor (HE-1) where the condensate was collected in the condensate trap (CT-1), and (ii) by the gas analyzer cooler (HE-2) with a condensate trap (CT-2) installed to prevent moisture or condensation in the gas analyzer. The dried product gas was sent to the online gas analyzer (GA) consisting of a Caldos27 thermal conductivity analyzer for measuring the hydrogen concentration (measurement ranges: 0–0.5 vol% and 0–100 vol%; output error (2σ): ≤0.5% of smallest measurement range span) and an Uras26 infrared photometer for measuring the $CO_2$, CO, and $CH_4$ concentration (measurement ranges: 0–10 vol% and 0–100 vol%; output error (2σ): ≤0.2% of span).

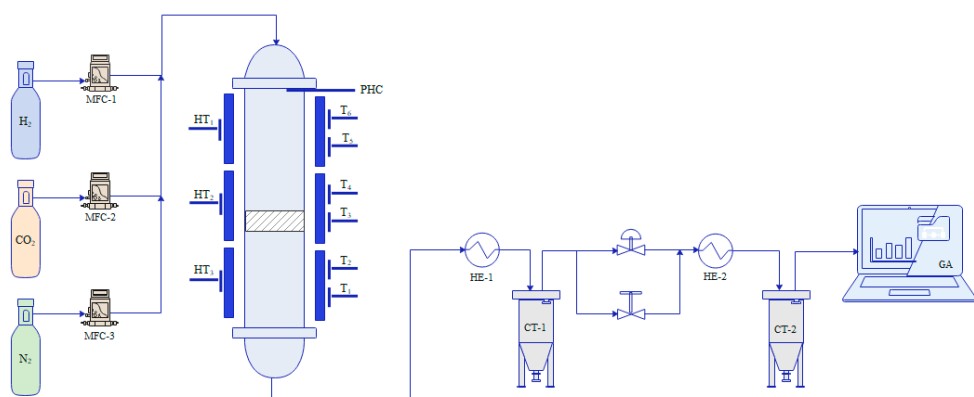

**Figure 11.** Experimental setup of the tubular reactor used for the $CO_2$ methanation experiments (MFC: mass flow controller, $T_1$–$T_6$: thermocouple inside the reactor tube, $HT_1$–$HT_3$, temperature measurement position in the middle of a heating zone, PHC: pre-heating coil, HE: heat exchanger, CT: condensate tank, BPR: back pressure regulator, GA: gas analyzer).

### 3.4.2. Experimental Procedure

$CO_2$ methanation experiments were performed at ambient pressure, feed gas flow rates of $8.02-14.66 \, \text{m}^3 \, \text{kg}^{-1} \, \text{h}^{-1}$ (STP), and a constant volumetric feed gas ratio ($H_2$:$CO_2$:$N_2$ = 56:14:30). According to Sommerbauer et al. [72], preliminary temperature scanning tests were conducted for each catalyst to evaluate the characteristic temperature effects of the respective catalyst regarding $CO_2$ conversion and methane selectivity before proceeding to

steady-state experiments. Therefore, the reaction temperature range was set to 548–623 K. 4 g of the catalyst sample was used for each experiment. It was placed in the middle of the reactor tube. Pure nitrogen was used for purging the system at ambient temperature and pressure, and the composition was checked by the online gas analyzer. Next, the feed gas stream ($H_2$:$CO_2$:$N_2$ = 56:14:30) with the designed feed gas flow rate was fed to the reactor while the heater was heated to the target temperature, which was measured at the end of the catalyst bed ($T_{cat} = T_3$). The gaseous products in the product gas stream were analyzed by the online gas analyzer after passing two heat exchangers for condensation of water. The dry gaseous product stream consisted of $N_2$, $CO$, $CO_2$, $CH_4$, and $H_2$ only. There were no other constituents present in the dry product gas (at concentrations exceeding 0.1 vol%).

The performance of the catalysts was evaluated with regard to $CO_2$ conversion ($X_{CO_2}$, Equation (14)) and $CH_4$ selectivity ($S_{CH_4}$, Equation (15)), with $c_{CO_2,0}$ being the initial concentration of $CO_2$ and $c_i$ being the concentration of any species i at the outlet of the reactor. The volumetric expansion coefficient $\varepsilon_{CO_2}$ was calculated by Equation (16), with $y_{CO_2,0}$ being the molar feed fraction of $CO_2$ and the stoichiometric coefficients of the methanation reactions a, b, c, and d ($CO_2$: a = −1, $H_2$: b = −4, $CH_4$: c = 1, and $H_2O$: d = 2). The concentrations of the species ($c_{N_2}$, $c_{H_2}$ and the product concentration $c_P$ for the products $CH_4$ and $CO$, respectively) were calculated with Equations (17)−(19).

$$X_{CO_2} = \frac{c_{CO_2,0} - c_{CO_2}}{c_{CO_2,0} + \varepsilon_{CO_2} \cdot c_{CO_2}} \tag{14}$$

$$S_{CH_4} = \frac{c_{CH_4}}{c_{CH_4} + c_{CO_2}} \tag{15}$$

$$\varepsilon_{CO_2} = y_{CO_2,0} \cdot \left( \frac{|c|}{|a|} + \frac{|d|}{|a|} - \frac{|b|}{|a|} - 1 \right) \tag{16}$$

$$c_{N_2} = \frac{c_{N_2,0}}{1 + \varepsilon_{CO_2} \cdot X_{CO_2}} \tag{17}$$

$$c_{H_2} = \frac{C_{H_2,0} - b \cdot X_{CO_2} \cdot c_{CO_2,0}}{1 + \varepsilon_{CO_2} \cdot X_{CO_2}} \tag{18}$$

$$c_P = \frac{(|c| \text{ or } |d|) \cdot X_{CO_2} \cdot c_{CO_2,0}}{1 + \varepsilon_{CO_2} \cdot X_{CO_2}} \tag{19}$$

### 3.4.3. Determination of the Chemical Composition of the Reduced Siderite Ore

The chemical composition of the reduced siderite ore was characterized in a five step analysis procedure at the Chair of Mineral Processing, Montanuniverstät Leoben:

(i)   Measurement of the weight increase in air (equaling the reactivity of the sample in air) and determination of the oxidation state by loss on ignition (LOI, fully oxidized and in a neutral atmosphere);

(ii)  Combustion analysis via the Leco method to determine the total (residual) carbon content;

(iii) Selective dissolution of metallic iron from iron oxides in bromine/methanol to determine elemental iron and for the determination of dissolved iron as $Fe^{II}$ via the Zimmermann–Reinhardt method;

(iv)  Digestion of the filter cake in boiling hydrochloric acid (HCl) to determine bivalent $Fe^{II}$ and trivalent $Fe^{III}$ iron using the Zimmermann–Reinhardt method;

(v)   X-ray diffraction of the residual elements.

The degree of metallization ($w_{met}$) is defined as the mass of elemental iron ($m_{Fe}^0$) with respect to the total mass of iron-bearing components in the reduced iron ore ($m_{Fe,tot}$).

## 4. Conclusions

In this study, the catalytic potential of hydrogen-reduced siderite ore, and Ni-based catalysts on reduced siderite ore and mixed reduced siderite ore/MgO support material was evaluated. It was shown that siderite ore that was reduced in a hydrogen atmosphere can act as a $CO_2$ hydrogenation catalyst but only causes low selectivity toward $CH_4$. Ni-based catalysts on reduced siderite ore only showed marginally higher catalytic activity than undoped reduced siderite ore. However, it was proven that MgO plays a fundamental role as a basic support material for Ni-based $CO_2$ methanation catalysts by drastically enhancing both $CO_2$ conversion and selectivity toward $CH_4$. When MgO was present in the support material, even fractions as low as 30% resulted in $CO_2$ conversions close to the thermodynamic equilibrium and $CH_4$ selectivities of at least 95% (mainly $\geq$ 99.9%). It could be shown that reduced siderite ore itself shows minor catalytic activity for $CO_2$ methanation but, in combination with MgO, has a clear synergistic effect as a support material for Ni-based catalysts giving access to highly efficient $CO_2$ methanation catalysts

**Author Contributions:** Conceptualization, S.L. and K.S.; methodology, S.L. and K.S.; validation, K.S.; formal analysis, K.S.; investigation, K.S.; resources, S.L.; data curation, K.S and S.S.; writing—original draft preparation, K.S. and S.L.; writing—review and editing, S.L. and C.A.H.; visualization, K.S.; supervision, S.L.; project administration, S.L.; funding acquisition, K.S. and S.L. All authors have read and agreed to the published version of the manuscript.

**Funding:** This research was funded by Zukunftsfonds Steiermark, grant number 1481. Open Access Funding by the Graz University of Technology.

**Data Availability Statement:** The data presented in this study are available on request from the corresponding author.

**Acknowledgments:** The authors acknowledge financial support from NAWI Graz. The analytical equipment data was supported by Michael Gostencnik, Department of Earth Sciences—NAWI Graz Geocenter, University of Graz. Special thanks go to Alfred Stadtschnitzer from VA Erzberg GmbH for providing the siderite ore. Publication was supported by TU Graz Open Access Publishing Fund.

**Conflicts of Interest:** The authors declare no conflicts of interest. The funders had no role in the design of the study; in the collection, analyses, or interpretation of data; in the writing of the manuscript; or in the decision to publish the results.

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
