# Peer review of "Reduced Siderite Ore Combined with Magnesium Oxide as Support Material for Ni-Based Catalysts; An Experimental Study on CO2 Methanation"

_catalysts, doi:10.3390/catal14030206_

Round 1

Reviewer 1 Report

Comments and Suggestions for Authors

In this study, the focus is on the research of carbon dioxide (CO2) methanation as a potential CO2 utilization technology. The emphasis is on the influence of nickel-based catalysts and support materials on catalytic performance, with particular attention to the role of magnesium oxide (MgO) as an effective support material. The results indicate that, at 648 K, the hydrogenation activity of CO2 on reduced siderite is comparable, regardless of the presence of nickel, but the methane selectivity is low. However, the introduction of MgO into the support material significantly enhances both the CO2 conversion rate and methane selectivity. The CO2 conversion rate approaches thermodynamic equilibrium, with methane selectivity exceeding 99%. This suggests that MgO, as a support material, can significantly improve the performance of nickel-based catalysts and effectively promote the CO2 methanation reaction. The study investigates the catalytic performance of nickel-based catalysts on reduced siderite with and without MgO. Therefore, I recommend acceptance after the authors provide additional content, clarify ambiguous points, and correct formatting issues.

1. Is there any side reaction during the CO2 + H2 methanation process in this article?

2. Why was the gas ratio chosen as H2:CO2:N2 = 56:14:30 during the catalyst evaluation process? We hope the authors provide an explanation.

3. Can the authors provide a comparison of the elemental composition of siderite before and after reduction, along with an explanation?

4. How to detect the proportion of different oxidation states of Fe elements in siderite after reduction?

5. Why does the increase in nickel content, as observed in Figure 4(b), lead to a decrease in the catalyst's selectivity for methane?

6. The author mentions in the paper that alkalinity affects the catalytic activity of the catalyst. Therefore, is there a noticeable difference in alkalinity between catalysts with different ratios of MgO and reduced siderite?

7. Why is there such a noticeable change in the intensity of the catalyst's XRD diffraction peaks in Figure 8 before and after the reaction? The author is requested to provide an explanation.

Comments on the Quality of English Language

Moderate editing of English language is required.

Author Response

Dear respected reviewer,

Thank you for your time and effort. Your valuable comments and productive suggestions helped to improve the quality of the manuscript a lot. The manuscript was checked and corrected by an English-editing service to enhance the quality of the English language. Furthermore, it was restructured according to your suggestions. Please find the detailed responses below and and the corrections highlighted (yellow highlight) in the re-submitted files.

Point-by-point response to Comments and Suggestions for Authors

1. Is there any side reaction during the CO2+ H2 methanation process in this article?

Ans: As described in the introduction, the reverse water-gas shift is the main side reaction for CO2 methanation. When the reaction temperature was increased, the CH4 selectivity decreased and CO was formed instead. This was described in more detail and added to the manuscript. (Page 7; Line 240)

2. Why was the gas ratio chosen as H2:CO2:N2= 56:14:30 during the catalyst evaluation process? We hope the authors provide an explanation.

Ans: This explanation also added to the manuscript. (Page 6; Line 183)

The feed gas ratio of CO2:H2 was 4:1. Inert nitrogen was added to the feed gas stream for balancing purposes (H2:CO2:N2 = 56:14:30). From literature, it can be deduced that the availability of adsorbed hydrogen is a limiting factor for the rate of reaction of CO2 methanation. Loder et al. investigated this effect and varied the H2:CO2 ratio in the feed gas from 3:1 to 5:1 [9]. As expected, the CO2 conversion rose with rising hydrogen concentration in the feed gas stream yielding a maximum CO2 conversion of 98% (equilibrium con-version: 99.8%) for the H2:CO2 ratio of 5:1. However, as most of the experiments in the study of Loder et al. were performed with a stoichiometric feed gas ratio of CO2:H2 = 4:1, for reasons of comparability, this ratio was also chosen in this study.

3. Can the authors provide a comparison of the elemental composition of siderite before and after reduction, along with an explanation?

Ans: The manuscript was restructured to make the difference in chemical composition of the reduced siderite ore more clear (former Table 2 was removed). The siderite ore mainly consists of carbonates – see manuscript. (Page 16-17; Line 484-489)

This siderite ore consists of the iron-bearing minerals siderite ((Fe0.83Mg0.11 Mn0.05Ca0.01)CO3) with substitutions of magnesium carbonate (MgCO3), calcium carbonate (CaCO3) and manganese carbonate (MnCO3), and ankerite ((Ca0.51Fe0.31Mg0.15Mn0.03)CO3). Furthermore, dolomite ((Ca,Mg)(CO3)2), calcite (CaCO3), quartz (SiO2), and muscovite (KAl2(Al-Si3O10)(OH)2) are present in this carbonaceous ore (Table 2).

When reduced with hydrogen CO2 is released and FeCO3 is converted to metallic iron and FeO. The concomitant carbonates are converted to their respective oxides.

4. How to detect the proportion of different oxidation states of Fe elements in siderite after reduction?

Ans: The chemical composition of the reduced siderite ore was characterized in a five step analyzing procedure. The procedure is described in section “3.4.3 Determination of the chemical composition of the reduced siderite ore” which was added to the manuscript. (Page 22; Line 636-651)

5. Why does the increase in nickel content, as observed in Figure 4(b), lead to a decrease in the catalyst's selectivity for methane?

Ans: An explanation was added: (Page 11; Line 347-351)

This suggests that in this range a maximum Ni load is present, which is advantageous in terms of methane selectivity. Furthermore, it must be noted that the production of catalysts with such a high Ni loading on reduced siderite ore is more difficult and more difficult to reproduce, which could also be reflected in the experimental results. Furthermore, XRD results revealed that with increasing Ni loading cohenite (Fi, Ni, CO)3C can be found. Iron carbide can occur during the reaction which is unstable. The iron carbide can cause ferrite and graphite formation (austenite = γ solid solution; ferrite = α solid solution) depending on the temperature. This solid solution could block the active sites of the catalyst and thus effect the CH4 selectivity.

6. The author mentions in the paper that alkalinity affects the catalytic activity of the catalyst. Therefore, is there a noticeable difference in alkalinity between catalysts with different ratios of MgO and reduced siderite?

Ans: Alkalinity was not measured, sorry for that.

7. Why is there such a noticeable change in the intensity of the catalyst's XRD diffraction peaks in Figure 7 before and after the reaction? The author is requested to provide an explanation.

Ans: An explanation was added in the scientific discussion. Moreover, the XRD peaks were marked in Figure 7 with explanation in the manuscript. (Page 15; Line 453-457)

Response to Comments on the Quality of English Language

Thank you for your responsing. We agree with the comment. Therefore, the manuscript was checked and corrected by an English-editing service to enhance the quality of the English language and the corrections highlighted in the re-submitted files.

Reviewer 2 Report

Comments and Suggestions for Authors

 1.       The topic of the article addresses the urgent and important challenges of CO2 reduction and sustainable management of natural resources through the use of catalytic processes, and is therefore in line with the main themes of the journal Catalysts.

2.       The paper presents interesting results on the use of natural siderite ores as components of new catalysts for CO2 methanation reactions. The authors pointed out an interesting direction for the further development of nickel- and iron-based catalysts by mixing with an acid-base specific component (MgO). In my opinion, the article can be accepted for publication, but some points need to be reconsidered.

3.       Please check the reaction equations, correct the relevant stoichiometric coefficients (e.g. reaction equations 4, 6,7,8,11).

4.       It is not clear what the composition of the catalysts was. Table 1 shows that the catalyst labelled "24 ..." contains 24.8 wt% NiO. While from the text of the article it can be understood that the catalyst contains 24 wt% Ni (not NiO). Furthermore, it is not clear whether the Ni content refers to the catalyst in calcined or reduced form directly introduced to the reactor. What was the Ni content of the reduced catalysts packed into the reactor.

5.       A detailed analysis of the XRD results could raise additional interesting issues. The position of the reflection lines corresponding to specific phases could be marked on the curves. It would be useful to determine the width of the peaks, especially for NiFe alloys, in order to discuss the size of crystallites with the catalytic results.

6.       It is not clear what exactly was the reason for the changes in catalyst activity with changing Ni content - changes in the number of active sites, changes in the nature of the Ni, NiFe active sites due to e.g. a change in Ni/Fe ratio, changes in redox properties? It was noted that the catalyst contained an alloy of (Fe,Ni,Co)3C. Please explain the presence of cobalt or the formation of pure NiFe carbide-like alloys, this could be an interesting point. It is difficult to agree unequivocally with the thesis presented in lines 402-403 that the activity of the reduced catalysts studied here was related to the absence of medium base sites and SMSI effects. Perhaps such a concept was valid for the refereed article, but neither the acid-base properties nor the SMSI effects were investigated here. Perhaps a detailed analysis of the results of XRD, electron microscopy or additional experimental studies would have been more helpful.

7.       Section 2.4.3 The same notation for catalysts as in the rest of the article should be used. It is difficult to agree with the conclusion (without the use of more sophisticated experimental techniques such as SEM-EDX or XPS) that the SEM images presented reveal the presence of species with exact chemical composition, i.e. NiO or FeO particles. The possibility of the presence of more complex phases is discussed in section 2.4.2.

8.       It was indicated that the introduction of MgO into the catalytic bed could significantly increase the activity of the catalysts. This is a very interesting aspect, but in my opinion insufficiently discussed with regard to the reaction mechanism, perhaps the possible effects of heat and mass transfer or the durability of the catalysts.

Author Response

Dear respected reviewer,

Thank you for your time and effort. Your valuable comments and productive suggestions helped to improve the quality of the manuscript a lot. The manuscript was checked and corrected by an English-editing service to enhance the quality of the English language. Furthermore, it was restructured according to your suggestions. Please find the detailed responses below and and the corrections highlighted (yellow highlight) in the re-submitted files.

  1. The topic of the article addresses the urgent and important challenges of CO2 reduction and sustainable management of natural resources through the use of catalytic processes, and is therefore in line with the main themes of the journal Catalysts.

Ans: Thank you for your kind comment.

      2. The paper presents interesting results on the use of natural siderite ores as components of new catalysts for CO2 methanation reactions. The authors pointed out an interesting direction for the further development of nickel- and iron-based catalysts by mixing with an acid-base specific component (MgO). In my opinion, the article can be accepted for publication, but some points need to be reconsidered.

Ans:  We agree with the comment. Therefore, we reconsider and restructure the manuscript as re-submitted files.

  1. Please check the reaction equations, correct the relevant stoichiometric coefficients (e.g. reaction equations 4, 6,7,8,11).

Ans: The reaction equations were checked and corrected.

Eqn(4) : FeCO3 + 0.25O2 → 0.5 Fe2O3 + CO

Eqn(6) : 4 (MgCO3) ∙ Mg(OH)2 ∙ 4 H2O → 5 MgO + 4 CO2 + 5 H2O

Eqn(7) : FeCO3 + (x+y+4z) H2 → FeO1-x + (1-y-z)CO2 + yCO + zCH4 + (x+y+2z)H2O

Eqn(8) : Ni(NO3)2 ∙ 6 H2O → NiO + 2NO2 + 0.5 O2 + 6 H2O

Eqn(11): Mg1-xNixO2 + H2 → [Ni]x*[Mg 1-x O] + H2O

  1. It is not clear what the composition of the catalysts was. Table 1 shows that the catalyst labelled "24 ..." contains 24.8 wt% NiO. While from the text of the article it can be understood that the catalyst contains 24 wt% Ni (not NiO). Furthermore, it is not clear whether the Ni content refers to the catalyst in calcined or reduced form directly introduced to the reactor. What was the Ni content of the reduced catalysts packed into the reactor.

Ans: It was tried to clarify this in the text. The Ni-loading is given as wt% of Ni (not NiO) (Page 14; Line 428-430)

  1. A detailed analysis of the XRD results could raise additional interesting issues. The position of the reflection lines corresponding to specific phases could be marked on the curves. It would be useful to determine the width of the peaks, especially for NiFe alloys, in order to discuss the size of crystallites with the catalytic results.

Ans: We improved the explanation about crystallites size of NiFe alloys and marked the peak of XRD in the Figures following the reviewer comments. (Page 15; Line 453)

  1. It is not clear what exactly was the reason for the changes in catalyst activity with changing Ni content - changes in the number of active sites, changes in the nature of the Ni, NiFe active sites due to e.g. a change in Ni/Fe ratio, changes in redox properties? It was noted that the catalyst contained an alloy of (Fe,Ni,Co)3C. Please explain the presence of cobalt or the formation of pure NiFe carbide-like alloys, this could be an interesting point. It is difficult to agree unequivocally with the thesis presented in lines 402-403 that the activity of the reduced catalysts studied here was related to the absence of medium base sites and SMSI effects. Perhaps such a concept was valid for the refereed article, but neither the acid-base properties nor the SMSI effects were investigated here. Perhaps a detailed analysis of the results of XRD, electron microscopy or additional experimental studies would have been more helpful.

Ans: The results were discussed more appropriately. The presence of significant quantities of cobalt can be excluded. The section regarding SMSI was removed. (Page 15; Line 453-457)

  1. Section 2.4.3 The same notation for catalysts as in the rest of the article should be used. It is difficult to agree with the conclusion (without the use of more sophisticated experimental techniques such as SEM-EDX or XPS) that the SEM images presented reveal the presence of species with exact chemical composition, i.e. NiO or FeO particles. The possibility of the presence of more complex phases is discussed in section 2.4.2.

Ans: That is absolutely correct. Regarding the limit of equipment, the EDX function of SEM equipment was not able to reveal the presence of the species of chemical composition.

  1. It was indicated that the introduction of MgO into the catalytic bed could significantly increase the activity of the catalysts. This is a very interesting aspect, but in my opinion insufficiently discussed with regard to the reaction mechanism, perhaps the possible effects of heat and mass transfer or the durability of the catalysts.

Ans: The results were discussed in more detail. (Page 22; Line 662-665)

Reviewer 3 Report

Comments and Suggestions for Authors

The manuscript titled “Reduced siderite ore combined with magnesium oxide as support material for Ni-based catalysts; an experimental study on CO2 methanation” presented Kamonrat Suksumrit, Christoph Hauzenberger , Srett Santitharangkun , and Susanne Lux.

  • The first part of Abstract looks as a part of Introduction (11-15). The current version of abstract should be revised. Abstract should content the key features of the manuscript.
  • Introduction is too broad and should be revised. It is recommended to sum up the observed articles to Table. And the novelty and aim of the study should be presented more clearly.
  • Line 188: Flow rate is in m3 kg-1 min-1, but in m3 kg-1 h-1 at figure 1. Actually, the flow rate is measured in sccm, slm, or cm3/min, etc. What “kg” means?
  • Figures should be colored. It is free of charge.
  • Could you discuss the observed decrease in the methane selectivity with increase of CO2 conversion for siderite reduced at 973K? What is the reason for this?
  • How was the part of reduced iron established? Line 230-231. The authors cited the early published data (source 26) where the metallization was evaluated according to thermogravimetric analyses of the mineral iron carbonate sample containing 80wt.% of siderite (siderite ore) after inert storage. In the manuscript the authors did not clarify what method was used to establish the fraction of iron cations in the reduced siderite ore after atmospheric storage.
  • Figure 3 should be converted to table or like figure 4. It is difficult to compare the presented data.
  • Discussion section has no any discussion of obtained result. It just repeats the information presented in Result section. For example, see lines 197-235 and 367-383.
  • Fig. 6 contents the data considered to CO2 methanation over 31%Ni/MgO. This catalyst does not discuss in the manuscript and appear only in this section and briefly mentioned in the preparation section.
  • Please, mark all phases found (lines 339-341) at the diffraction patterns (Fig. 7).

The aim of the study was to evaluate of effect of siderite ore and siderite/MgO as support material for Ni-based catalysts, but authors conclude that “that at ident Ni loading, Ni/siderite(reduced)/MgO (≥ 30 % MgO) and Ni/MgO catalysts show comparable catalytic performance with CO2 conversions close to the thermodynamic equilibrium and high CH4 selectivity ≥ 99.9 %”, In the other words, that the more MgO is the better for the CO2 methanation. And MgO play main role. The manuscript has a lack of discussion of obtained data. The discussion should be extended. The reduced siderite ore as support should be highlighted.

Comments on the Quality of English Language

English demands some corrections.

Author Response

Dear respected reviewer,

Thank you for your time and effort. Your valuable comments and productive suggestions helped to improve the quality of the manuscript a lot. The manuscript was checked and corrected by an English-editing service to enhance the quality of the English language. Furthermore, it was restructured according to your suggestions. Please find the detailed responses below and and the corrections highlighted (yellow highlight) in the re-submitted files.

  • The first part of Abstract looks as a part of Introduction (11-15). The current version of abstract should be revised. Abstract should content the key features of the manuscript.

Ans: The Abstract was rewritten to contain the key features of the manuscript. (Page 1; Line11-16)

  • Introduction is too broad and should be revised. It is recommended to sum up the observed articles to Table. And the novelty and aim of the study should be presented more clearly.

Ans: The Introduction was revised and shortened, and a Table was added. The aim of the study was presented more clearly. (Page 6 with yellow highlight)

  • Line 188: Flow rate is in m3 kg-1 min-1, but in m3 kg-1 h-1 at figure 1. Actually, the flow rate is measured in sccm, slm, or cm3/min, etc. What “kg” means?

Ans: The unit of the flow rate was changed so that it is uniformly used in the text and in Figure 1. The flow rate unit is m3 kg-1 h-1 referring to 1 kg of catalyst. In fact there was a typo when min was used instead of h. (mark in yellow highlight)

  • Figures should be colored. It is free of charge.

Ans: We changed most of the Figures to color for easier understanding. (Figure in page 8-13,15,19.21)

  • Could you discuss the observed decrease in the methane selectivity with increase of CO2 conversion for siderite reduced at 973K? What is the reason for this?

Ans: We added a more detailed discussion: (Page 7; Line 233-248)

Since the effect of increasing methane selectivity with increasing feed gas flow rate and thus lower residence time was only observed at the lowest reaction temperature of 548 K and only the methane selectivity at a feed gas flow rate of 8.02 m3 kg−1 h−1 differed from the one at 11.32 and 14.66 m3 kg−1 h−1, respectively, it is assumed that this value should rather be regarded as an outlier.

As the reaction temperature increased, the CH4 selectivity decreased. CO was formed instead, showing that the reverse water-gas shift reaction was the dominant reaction. This is consistent with the findings of Lux et al. who investigated the direct reduction process of siderite ore with hydrogen, called reductive calination in the study, with the aim of optimizing the process parameters to maximize the methane yield. Since it can be assumed that CO2 is released from the iron carbonate during direct reduction with hydrogen in a first step and is further reduced to methane in a subsequent catalytic step, the results are directly transferable. As expected, in their study, it was proven that methane formation is favored at low temperature and increased pressure, whereas the formation of CO is favored at high temperature and low pressure [61].

  • How was the part of reduced iron established? Line 230-231. The authors cited the early published data (source 26) where the metallization was evaluated according to thermogravimetric analyses of the mineral iron carbonate sample containing 80wt.% of siderite (siderite ore) after inert storage. In the manuscript the authors did not clarify what method was used to establish the fraction of iron cations in the reduced siderite ore after atmospheric storage.

Ans: The chemical composition of the reduced siderite ore was characterized in a five step analyzing procedure. The procedure is described in section “3.4.3 Determination of the chemical composition of the reduced siderite ore” which was added to the manuscript. (Page 22; Line 636-651)

  • Figure 3 should be converted to table or like figure 4. It is difficult to compare the presented data.

Ans: Figure 3 was modified and changed to color for easier understanding. (Page 10)

  • Discussion section has no any discussion of obtained result. It just repeats the information presented in Result section. For example, see lines 197-235 and 367-383.

Ans: Results and discussion were combined in one section named “Results and Discussion” in order to prevent repetition.

  • Fig. 6 contents the data considered to CO2 methanation over 31%Ni/MgO. This catalyst does not discuss in the manuscript and appear only in this section and briefly mentioned in the preparation section.

Ans: We clarified that in the text. (Page 6; Line 194-197 and Page 11; Line 358-359)

  • Please, mark all phases found (lines 339-341) at the diffraction patterns (Fig. 7).

Ans: We improved the XRD-picture in Figure 7 following the reviewer comments. (Page 15)

  • The aim of the study was to evaluate of effect of siderite ore and siderite/MgO as support material for Ni-based catalysts, but authors conclude that “that at ident Ni loading, Ni/siderite(reduced)/MgO (≥ 30 % MgO) and Ni/MgO catalysts show comparable catalytic performance with CO2 conversions close to the thermodynamic equilibrium and high CH4 selectivity ≥ 99.9 %”, In the other words, that themore MgO is the better for the CO2 methanation. And MgO play main role. The manuscript has a lack of discussion of obtained data. The discussion should be extended. The reduced siderite ore as support should be highlighted.

Ans: A more detailed discussion was added. (page 22' Line 662-665)

Response to Comments on the Quality of English Language

Thank you for your responsing. We agree with the comment. Therefore, the manuscript was checked and corrected by an English-editing service to enhance the quality of the English language and the corrections highlighted in the re-submitted files.

Round 2

Reviewer 3 Report

Comments and Suggestions for Authors

The authors have been revised the manuscript, answered questions and comments. The manuscript could be accepted for publication.

Comments on the Quality of English Language

Minor editing of English language required